



# Coastal processes modify projections of some climate-driven stressors in the California Current System

**Samantha A. Siedlecki[1][*], Darren Pilcher[2], Evan M. Howard[3], Curtis Deutsch[3], Parker MacCready[3], Emily L. Norton[2], Hartmut Frenzel[3], Jan Newton[4], Richard A. Feely[5], Simone R. Alin[5], and Terrie Klinger[6]**

[1]Department of Marine Sciences, University of Connecticut, Groton, CT 06340, USA
[2]CICOES, University of Washington, Seattle, WA 98195, USA.
[3]School of Oceanography, University of Washington, Seattle, WA 98195, USA.
[4]Applied Physics Laboratory, University of Washington, Seattle, WA 98105, USA.
[5]NOAA Pacific Marine Environmental Lab (PMEL), Seattle, WA 98115, USA.
[6]School of Marine Environment and Affairs, University of Washington, Seattle, WA 98105, USA.

*Correspondence to*: Samantha Siedlecki (samantha.siedlecki@uconn.edu)

**Abstract.** Global projections for ocean conditions in 2100 predict that the North Pacific will experience some of the largest changes. Coastal processes that drive variability in the region can alter these projected changes, but are poorly resolved by global coarse resolution models. We quantify the degree to which local processes modify biogeochemical changes in the eastern boundary California Current System (CCS) using multi-model regionally downscaled climate projections of multiple climate-associated stressors (temperature, $O_2$, pH, $\Omega$, and $CO_2$). The downscaled projections predict changes consistent with the directional change from the global projections for the same emissions scenario. However, the magnitude and spatial variability of projected changes are modified in the downscaled projections for carbon variables. Future changes in $pCO_2$ and surface $\Omega$ are amplified while changes in pH and upper 200 meter $\Omega$ are dampened relative to the projected change in global models. Within the CCS, differences in global and downscaled climate stressors are spatially variable, and the northern CCS experiences the most intense modification. These projected changes are consistent with source waters lower in oxygen, higher in nutrients, and in combination with solubility-driven changes, altered future upwelled waters in the CCS. The results presented here suggest coastal process resolving projections are necessary for adequate representation of the magnitude of projected change in pH and carbon stressors in the CCS.

## 1 Introduction

Greenhouse gas emissions have imparted large physical and biogeochemical modifications on the world's oceans (Friedlingstein et al., 2019; Gattuso et al., 2015; Le Quéré et al., 2018). The oceans have become warmer and stratification patterns have been modified (Talley et al., 2016). These changes are occurring in tandem with biogeochemical alterations, including $O_2$ declines, productivity changes, and increased dissolved inorganic carbon content due to the uptake of anthropogenic carbon dioxide (Doney et al., 2009, 2020; Feely et al. 2004, 2009). The ocean uptake of anthropogenic carbon dioxide influences the ocean's buffering capacity, reduces calcium carbonate saturation states ($\Omega$), and lowers pH causing a



shift towards more acidity, commonly termed ocean acidification (Caldeira and Wickett, 2003; Doney et al., 2009; Feely et al., 2004, 2009; Orr et al., 2005; Sabine et al., 2004). These changing ocean conditions are occurring in both open-ocean and

coastal environments, where they have the potential to impact marine organisms and ecosystems individually (Doney et al., 2012, 2020; Gattuso et al., 2015) and as interactive multi stressors (Howard et al., 2020; Portner et al., 2004; 2007). Big changes are happening in the ocean, but there are reasons to believe that global trends may not accurately represent what happens in coastal regions.

The majority of coastal areas have experienced significant increases in sea-surface temperature (SST) at a rate that exceeds the global average (Hartmann et al., 2013; Lima and Wethey, 2012). In contrast to most other large marine ecosystems, the coastal SST in the California Current System (CCS) has decreased over the past three decades (Lima and Wethey, 2012). Within the CCS over a shorter time period, in situ observations from Monterey Bay in the central CCS indicate a decadal increase in SST (Chavez et al., 2017). These results suggest global projections and trends are poor indicators of future change

in SST in the CCS and that spatial variability of that change within the CCS is also possible.

As the oceans warm, they lose $O_2$ because the solubility of $O_2$ decreases with increasing temperature. However, direct solubility effects only partially explain the $O_2$ decline (Bopp et al., 2013; Breitburg et al., 2018). Warming impacts $O_2$ in other ways, for example by raising organismal metabolic rates and accelerating $O_2$ consumption, and by increasing water column stratification

and thus reducing mixing and ventilation (Breitburg et al., 2018; Deutsch et al., 2006). In coastal waters, hypoxic thresholds are more often reached than in the open ocean because of eutrophication and other local processes, such as sediment $O_2$ demand (Diaz and Rosenberg, 2008; Rabalais et al., 2010; Siedlecki et al. 2015). In the CCS, hypoxia already occurs regularly on the continental shelf (Adams et al., 2013; Connolly et al., 2010; Chan et al., 2008; Grantham et al., 2004), and continental slope water $O_2$ concentrations have been steadily declining for the past several decades (Bograd et al. 2008; Chavez et al., 2017;

Deutsch et al., 2011; 2014; Pierce et al., 2012).

Atmospheric carbon dioxide has increased at a rate of 1-2 ppm per year (Friedlingstein et al., 2019; Le Quéré et al., 2018), and surface waters in the open ocean have effectively kept pace with the rising atmospheric concentrations over the last 30 years. Recently, the partial pressure of carbon dioxide ($pCO_2$) in coastal shelf waters has been shown in some places to lag the rise

in atmospheric $CO_2$, unlike in the open ocean, which implies a tendency for enhanced shelf uptake of atmospheric $CO_2$, with substantial regional variability (Laruelle et al., 2018). One example of regional variability is found in the CCS: over the past 30 years, the $CO_2$ content of waters off the U.S. West Coast near Monterey Bay, CA, has increased at a rate greater than that observed in the open oceans (Chavez et al., 2017). As a result, pH in Monterey Bay is on average 0.01 units lower than open ocean measurements at the Hawaii Ocean Time-series due to the upwelling process, and is also declining at a slightly faster

rate (Chavez et al., 2017). The enhanced uptake of $CO_2$ over the CCS shelf amplifies the rate of acidification compared to global rates.





Local processes such as upwelling, freshwater delivery, eutrophication, water column metabolism, and sediment interactions drive biogeochemical variability on regional scales (Cai et al., 2011; Feely et al., 2008; 2016; 2018; Pilcher et al., 2018; Qi et

al., 2017; Siedlecki et al., 2017). In the CCS, winds are critical for upwelling variability and are projected to strengthen in response to global warming (Bakun, 1990; Garcia-Reyes et al., 2015; Wang et al., 2015; Rykaczewski et al., 2015; Sydeman et al., 2014). The increased delivery of $O_2$-depleted, carbon-rich waters, with enhanced nutrients and increased productivity has been projected for the CCS with a global simulation (Rykaczewski and Dunne, 2010). High-resolution projections for the CCS reinforced these findings (Dussin et al. 2019; Xiu et al., 2018), but projected that the impact of these altered conditions

on productivity varied across the CCS with an increase in the north and a decrease in the south (Xiu et al., 2018), while productivity was identified as driving the biggest change in hypoxia in the region (Dussin et al., 2019). Global projections have coarse spatial resolution, often having only one or two grid cells for the continental shelf, and thus cannot resolve most of the local processes responsible for these observed coastal differences.

In this paper, we focus on the CCS and its known vulnerabilities to climate change by forcing regional models with the Coupled Model Intercomparison Project 5 (CMIP5, Taylor et al., 2012) simulations. We produce multi-model regionally downscaled climate projections of multiple climate-associated stressors (temperature, $O_2$, pH, $\Omega$, and $CO_2$) that resolve coastal processes to create 100-year projections at resolutions of 12-km and 1.5-km in the northern CCS (N-CCS). The downscaled projections are referred to as "resolving coastal processes" because the historical simulations have been shown to represent observed

coastal shelf variability. The model evaluation, provided as supplemental material and available in other sources (Davis et al. 2014; Deutsch et al. in review; Giddings et al. 2014; Siedlecki et al. 2015), indicate the downscaled models perform better spatially and temporally (seasonal and interannual) than the global models for temperature, oxygen and carbon variables. First, we quantify the surface to 200-meter depth averaged, sea surface, and bottom condition changes for the climate stressors projected for 2100 at all resolutions (global, 12-km, and 1.5-km). Next, we use the multi-model ensemble to determine the

degree to which climate associated stressors are modified relative to global model projections of the CCS considering this signal both spatially, where the models overlap, as well as in different regions of the water column representative of different habitats. Finally, we interpret our results in the context of previous projections for the CCS and suggest drivers of the amplification in the downscaled projections by systematically comparing the projected changes in the winds, source waters, upwelling strengths, and coastal processes in each model system.

**2 Materials and Methods**



## 2.1 Model Descriptions

The downscaled regional modeling frameworks both employ the Regional Ocean Modeling System (ROMS, Shchepetkin and McWilliams, 2005). The regional models are forced with realistic atmospheric and ocean boundary conditions to make hindcast simulations as well as future projections. The model domains are shown in Figure 1.

~1-degree models: These include the CMIP model fields/global scale model. We focus here on only the representative concentration pathway 8.5 (RCP 8.5) from the Earth system models (ESMs) that make up the CMIP5 modeling framework. The CMIP5 simulations include biogeochemical components described in Bopp et al. (2013).

12-km model: The mid-resolution (12-km) ROMS-based simulation of the CCS is configured for a domain that extends along the North American west coast from 25 deg N to 60 deg N and described in more detail in Howard et al. (in review). A curvilinear grid is used in the horizontal with close-to-uniform 12-km horizontal resolution and 33 s-coordinate (terrain-following vertical) levels. The biogeochemical model is detailed in Deutsch et al. (2020), and follows Moore et al. (2004). The model has skillfully simulated the recent interannual to interdecadal biogeochemical variability in the CCS (Howard et al. in review), and a similar model setup forced with data-assimilated forcing skillfully simulated $O_2$ variability in the CCS over the last two decades (Durski et al., 2017).

1.5-km model: The highest resolution (1.5-km) simulations of the N-CCS rely on a modeling framework developed by the University of Washington Coastal Modeling Group optimized for the Pacific Northwest "Cascadia" region. The Cascadia model domain encompasses the inland waters of the Salish Sea, coastal waters of the N-CCS (Fig. 1), and includes freshwater and tidal forcing. The grid has a horizontal resolution of 1.5-km on the shelf, 4.5 km far offshore, and 40 s-coordinate (terrain-following vertical) levels, with enhanced bottom and surface resolution. The Cascadia model does not simulate biogeochemistry within the Salish Sea, but yields realistic nitrate outflow from the Strait of Juan de Fuca to the outer coast shelf (Davis et al., 2014). Hindcast experiments from 2004 to 2007 were extensively validated and exhibited skill on all regions of the shelf (Davis et al., 2014; Giddings et al., 2014; Siedlecki et al., 2015).

## 2.2 Model Metrics

Carbon variables (e.g. pH, $pCO_2$, and $\Omega$) were computed using model output of dissolved inorganic carbon (DIC), total alkalinity (TA), temperature, and salinity and routines based on the standard OCMIP carbonate chemistry adapted from earlier studies (Orr et al., 2005) using CO2SYS (Lewis and Wallace, 1998). The total pH scale is used for pH throughout.

To compute model means and inter-model comparisons, first a climatological year was generated for each model grid cell, using the 2002-2004 years for the 12km and 1.5km regional models. Then, annual average values for each cell were calculated from the climatology. Finally, spatially-weighted means were calculated from the annual average values. For comparisons between mode resolutions, the coarser model was interpolated to the higher resolution model grid. For example, the global values were interpolated onto the 12-km grid prior to averaging the fields within the CCS. Surface conditions were drawn from the surface vertical layer for each simulation. Depth-averaged ocean conditions were calculated over the upper 200 m for



all simulations. Where water depth was shallower than 200 m, the entire water column was averaged. For the bottom

comparisons, the global simulations have a very different bathymetry than the downscaled simulations because of their coarse resolution, consequently the regional averaging efforts reported here as bottom conditions were isolated to the 0-500 meters depth interval only, to ensure that the global model resolved that depth interval. We also report the downscaled values over the shelf only, limiting the determination of metrics out to the 200 meter isobath.

Upwelling intensity – To estimate upwelling, several metrics were employed. The first two rely on the intensity of the winds (e.g. Cumulative Upwelling Index, CUI (Schwing et al., 1996); 8-day wind stress (Austin and Barth, 2002)), the third and fourth rely on measures in the water column itself and are referred to as the Coastal Upwelling Transport Index (CUTI) and Biologically Effective Upwelling Transport Index (BEUTI) (Jacox et al., 2018). The wind-based metrics, CUI and the 8-day, are the same for both downscaled simulations, but CUTI and BEUTI are specific to each ocean model as they are calculated

based on ocean measures like vertical transport and nitrate concentrations. CUTI and BEUTI were integrated over bins of 0.5-degree latitude spanning 0-50 km offshore.

### 2.3 Future Forcing

To generate future downscaled projections, the global CMIP5 simulations were used to force regional simulations. The methods employed are outlined below.


The 12-km historical simulation forcing is described in Renault et al (in review) and the companion paper, Deutsch et al. (in review). The 12-km projection was forced by adding a monthly climatological difference between CMIP5 RCP 8.5 scenario forcing and the historical run forcing, averaged over 2071-2100 and 1971-2000, respectively (Howard et al., in review). This is done for all variables that influence the surface energy budget including net downward shortwave and longwave radiation,

10-m wind speed (u and v component), air temperature, and specific humidity. CMIP5 models are from GFDL (ESM2M), IPSL (CM5A-LR), Hadley (GEM2-ES), MPI (ESM-LR), NCAR (CESM1(BGC)). A total of six RCP 8.5 scenario runs were conducted: one for each individual CMIP5 model realization, and a final run using the five-member ensemble mean forcing. For this manuscript, we report the output from this final, ensemble mean-forced scenario. However, the output from the five individual CMIP5 model realizations were used to calculate the standard deviation values across the ensemble spread reported

in Table 1. Initial and boundary conditions had the same kind of centennial trend addition for temperature, salinity, and all biogeochemical tracers ($O_2$, nitrate, phosphate, silica, iron, dissolved inorganic carbon, alkalinity).

The 1.5-km projection was forced using the open ocean boundary conditions and atmospheric forcing from the 12-km regional simulation described above (Howard et al., in review). The boundary conditions included biogeochemical fields from the 12-

km model. Because the ecosystem model (BEC) in the 12-km parent grid has more variables than the Cascadia simulation, some of the variables were merged. Specifically, the phytoplankton fields were added together, and the nutrients (ammonia





and nitrate) were summed into one nitrogen field. To ensure no biogeochemical model drift between the nested 1.5-km simulation and the 12-km simulation, 2007 with one year of spin up was compared against observations from the region (Fig. 2). The 1.5-km biogeochemical model skill was similar to the original model runs previously published (Davis et al. 2014;

Giddings et al. 2014; Siedlecki et al. 2015) for 2007 (Fig. 2) without any significant drift in time. Temperature and salinity both experienced a significant bias in the upper 200 meters in this configuration, unlike the previously published model runs (Fig. 2; temperature RMSE 2.43; salinity RMSE 0.627). As we are focused on differences between the modern and the future and we expect the bias to remain the same, we do not bias correct the forcing here.

For the future conditions, atmospheric $CO_2$ concentration (800 ppm), and future atmosphere and ocean forcing from the 12-km runs drove the Cascadia simulations. The river forcing was approximated by altering the timing of the freshet of the 2007 forcing earlier in the year by two months. This is in line with some historical analyses from the N-CCS region (Riche et al., 2014) as well as some projections of future hydrological conditions for the Fraser River (Morrison et al., 2002). Both of these results suggest that the total precipitation will remain the same, but the increase of rain and decrease in snowpack will shift the

freshet earlier. The river TA was not altered from historical forcing, but the rivers equilibrate with the future atmospheric $CO_2$ concentration (800 ppm).

### 2.4 Future Change

For each resolution, simulations were run for a number of years in a base/present state and then compared to a future simulation. The change is the difference between the future and base/present state, representing a 100-year anomaly due to climate forcing.

The historical/base state for the CMIP5 runs was computed from an ensemble mean spanning 1971-2000. For the 12-km simulation, the base/present state spanned 1994-2007. For the 1.5-km simulation, the model was spun up for one year (2001) and then the base/present state spanned 2002-2004. Each level of resolution entails additional computer resource costs, which is part of the appeal of large-scale simulations. The CMIP5 future runs consist of a thirty-year mean spanning 2071-2100. The 12-km model is a late 21st century run spanning 2085-2100. The 1.5-km model is also late 21st century spanning 2094-2096.

Comparisons between the 12- and 1.5-km-resolution simulations were made with the same year span despite runs existing for a broader range of years for the 12-km simulation. The global model ensemble average results represent a 30 year climatology.

### 2.5 Modification

The range of the five ensemble members which forced the 12-km projections is used to bound the potential futures expected. When the differences provided in Table 1 between the mean downscaled conditions for a region of the CCS (CCS-wide,

columns B-C or Cascadia/N-CCS, columns E-G) and the ensemble spread quantified from the 12-km model projections (columns D and H) both exceed the 1 degree model projected change (columns B and E), those regions of the CCS are projected to undergo amplified change. The converse is referred to as dampening. The ensemble spread is provided in Table 1 as the range of the 5-member model spread of the annual average results for the 12-km model projections (columns D and H). The



direction with which each variable described above was amplified relative to the global models is highlighted using pink
(amplified) and blue (dampened) colors in Table 1.

## 3 Results

Model projections of climate-driven stressor variables in each downscaled projection were compiled for the CCS for three
depths (200 m averaged, surface, and bottom < 500 m; Table 1; Fig. 3-5). The global average changes for many of these
variables are different from the 1 degree model projected values for the CCS region, but in only a few cases does this difference
fall outside of the ensemble spread reported in column D and H of Table 1 (i.e., the signal is amplified or dampened). For each
variable and depth, the change between the base state and the projected state is described below and evaluated within the
context of the ensemble spread. 1.2.1 Subsubsection (as Heading 3).

### 3.1 Temperature

The surface to 200-meter depth-averaged temperatures at all model resolutions is consistently warmer in the future CCS, both
CCS-wide (1.63 and 1.81 degrees C in the 1 degree and 12-km models, respectively, Table 1, columns B and C), and within
the N-CCS (1.94 to 2.32 degrees C across the three model resolutions; Figure 3a; Table 1 Columns E-G). The largest
temperature increase, nearly 3 degrees C, occurs on the shelf in both regions of the projections (Fig. 3, Table 1, asterisks).
Moreover, the Washington shelf experiences the largest projected differences in the 1.5-km projection (2.32 degrees C, Fig.
3). The 1-degree model projected increase for the CCS (1.63 degrees C) and the N-CCS (2.21 degrees C) falls within the range
of warming from the downscaled projections (CCS: 1.35 to 2.27 degrees C; N-CCS: 1.43 to 2.60 degrees C, Columns D and
H in Table 1). In both regions of the CCS, the differences between the models are smaller than the range of the 12-km ensemble
(Table 1).

At the surface, the SST is warmer in the future in all projections. Spatially, the SST increases most offshore, and increases
least near the coast in all simulations of this eastern boundary upwelling system, as a result of upwelling (Fig. 3b). The 1.5 km
model projects slightly smaller increases in SST than the 1-degree model or the 12-km model. However, the 1-degree models
project SST increases CCS-wide (3.12 degrees C) and in the N-CCS (3.15 degrees C), which fall within the range of SST
projections from the 12-km ensemble of downscaled projections (CCS: 2.53 to 4.07 degrees C; N-CCS: 2.34 to 4.24 degrees
C, Columns D and H in Table 1).

At the bottom, the temperature increases the most near the coast. The abyssal regions show little to no change in temperature
(Fig. 3c). The shallowest regions of the 1.5-km simulation experience the largest warming - nearly 3 degrees C. In the coastal
process-resolving downscaled projections, the projected bottom temperature change is greater (1.70-1.78 degrees C, 2.05
degrees C) than the global projections (1.34-1.65 degrees C). However, the 1-degree model projected increases for the whole





CCS (1.65 degrees C) fall within the range of temperature projections from the ensemble of 12-km projections (CCS: 1.39 to 2.17 degrees C, Column D in Table 1), while the N-CCS (1.34 degrees C) is lower than the range of temperature projections from the ensemble of 12-km projections (N-CCS: 1.40 to 2.30 degrees C, Columns H in Table 1).

**3.2 Oxygen**

Annual depth-averaged $O_2$ concentrations at all model resolutions consistently decrease in the future compared with the base
state, but the magnitude of the decrease is slightly more severe on average in the downscaled projections (Table 1; Fig. 3). The spatial variability within the CCS region, with more severe declines occurring in the N-CCS, is consistent across models but varies in magnitude. The 1-degree model projected decrease for the CCS (-0.52 ml/l) and the N-CCS (-0.56 ml/l) fall within the range of the ensemble of projections from the 12-km model (Table 1; CCS: -0.50 to -0.70 ml/l; N-CCS: -0.44 to -0.94 ml/l, Columns D and H in Table 1).

At the surface, $O_2$ declines in all projections, and the degree of change is similar across resolutions (Table 1; Fig. 3) consistent with the solubility effect from surface temperatures in each simulation. The 1-degree model projected decline falls within the spread ensemble of projections from the 12-km model for the CCS.

The bottom $O_2$ concentration declines in all projections near the coast (Table 1; Fig. 3). The range of change in bottom $O_2$ on the shelf in the 1.5-km projection varies by a factor of two, with the most extreme changes occurring on the outer shelf and in pockets known to experience persistent hypoxia in the present ocean - e.g. near Cape Elizabeth, south of Heceta Bank, and within the region associated with the Juan de Fuca Eddy (Siedlecki et al., 2015). The 1 degree model projected decrease for the CCS (-0.43 ml/l) and the N-CCS (-0.63 ml/l) falls within the ensemble range of the ensemble of bottom $O_2$ concentration
projections from the 12-km model (CCS: -0.39 to -0.73 ml/l; N-CCS: -0.37 to -0.96 ml/l, Columns D and H in Table 1). When the bottom is restricted to the shelf in the downscaled simulations (< 200 m isobath; values with asterisk (*) in Table 1), this decrease is more severe but still does not fall outside the range from the comparable depth range of the 1-degree model projection (<500 m).

**3.3 pCO2**

All model projections of $pCO_2$ consistently increase, with larger increases in the downscaled projections than in the global projection (Table 1; Fig. 4). The spatial variability within the CCS region differs across resolutions. All projections show an onshore-offshore trend in $pCO_2$ with smaller changes closer to the coast and larger changes offshore. In the coastal process-resolving downscaled projections, the projected depth-averaged change in $pCO_2$ increases, and the gradient between the nearshore and offshore intensifies. The 1-degree model projected increase for the CCS (492 µatm) and the N-CCS (527 µatm)
falls below the ensemble range of downscaled $pCO_2$ projections from the 12-km model (CCS: 682-824 µatm; N-CCS: 683 to 1044 µatm, Columns D and H in Table 1).





At the surface, future $pCO_2$ consistently increases in all projections, but varies widely across resolutions (Table 1; Fig. 4). In the downscaled projections, most upwelling areas experience a smaller increase in surface $pCO_2$ than offshore waters. In the 1.5-km projection, regions near the coast of Oregon show the largest surface $pCO_2$ differences between the base and future states, while the region associated with the Columbia River plume shows a much smaller change. Overall, the inclusion of coastal processes contributes to the spatial patterns and magnitudes of projected changes. Consistent with the subsurface signal, the 1-degree model projected increase for the CCS (392 µatm) and the N-CCS (365 µatm) fall below the ensemble range of the downscaled projections from the 12-km model (CCS: 432 to 436 µatm; N-CCS: 425 to 435 µatm).

At the bottom, $pCO_2$ is consistently higher in all projections, with varying magnitude across the model resolutions (Table 1; Fig. 4). The range of change in bottom $pCO_2$ on the shelf of the 1.5-km projection varies widely (600-1200 µatm), with the most extreme changes occurring on the outer shelf and in pockets known to experience persistent hypoxia at present. The 1-degree model projected change for the CCS (505 µatm) and the N-CCS (592 µatm) fall below the ensemble range of projections from the 12-km downscaled model (CCS: 663 to 903 µatm; N-CCS: 675 to 1164 µatm).

### 3.4 pH

The pH averaged over 200-meter depths for all model resolutions consistently decreases, and change is less severe than the global models project for the CCS region (Table 1; Fig. 4). In the 12-km simulation, a slightly smaller pH change (~ -0.26) is observed in the southern CCS than the entire CCS, within the influence of coastal upwelling. In the 1.5-km simulation, the regions of largest decline are on the outer shelf and patches of the Oregon shelf. In the downscaled projections, the depth-averaged 200-meter pH is lower than the 1-degree model projected change in the North and greater than the global change CCS -wide. The 1-degree model projected decrease for the CCS (-0.321) and the N-CCS (-0.332) fall within the ensemble range of projections for the downscaled 12-km model (CCS: -0.309 to -0.353; N-CCS: -0.252 to -0.343).

At the surface, the pH consistently decreases in all projections, and is less severe a decrease in the downscaled projections than for the same region in the global model (Table 1; Fig. 4). The 1-degree model projected decrease for the CCS (-0.319) and the N-CCS (-0.343) is larger than the ensemble range of projections for the downscaled 12-km model (CCS: -0.284 to -0.286; N-CCS: -0.294 to -0.300).

At the bottom, the pH decreases on the shelf in all projections (Table 1; Fig. 4). In the 1.5-km resolution model, the projected conditions show spatial variability on the shelf that is not apparent in the coarser models. The most severe changes in bottom pH correspond with regions that experience the largest changes in bottom $O_2$. The downscaled projections indicate decreases in annual average bottom (< 500 m) pH, but spatial variability exists on the shelves in the coastal process-resolving simulations. This difference between the 1 degree and downscaled simulations is even greater on the shelves (indicated as the starred values





in Table 1). The 1-degree model projected decline for the CCS (-0.286) falls within the ensemble range for the downscaled 12-km model (CCS: -0.231 to -0.287) while the N-CCS 1-degree model projection (-0.333) is larger than the ensemble range of projections (N-CCS: -0.206 to -0.310) for the downscaled 12-km model.

**3.5 Ω**

Projections of $\Omega_{arag}$ and $\Omega_{calcite}$ averaged over 200 m depths at all model resolutions consistently decrease in the future, but the
magnitude of decrease is usually greater in the downscaled projections (Table 1; Fig. 5). The projected difference is greater on the shelf than offshore, but this gradient is weaker in the 1.5-km projection. The 1 degree model projected decline in $\Omega_{arag}$ for the CCS (-0.71) falls within the ensemble range of projections for the downscaled 12-km model for the CCS (-0.65 to -0.75). The 1 degree projected decline in $\Omega_{arag}$ for the N-CCS (-0.62) is larger than the ensemble range of projections and larger than the ensemble range of projections for the N-CCS (-0.37 to -0.51) in the downscaled 12-km model. The same patterns are true
for the $\Omega_{calcite}$ averaged over 200 m depths (Table 1).

At the surface, Ω consistently decreases in all projections (Table 1; Fig. 5). Spatially, the 12-km projection shows the largest declines in surface Ω in the southern domain with little gradient between the shelf and offshore. At the 1.5-km resolution, the N-CCS projected declines are lowest offshore, and the declines are even larger than the 12-km projected changes even when
considering the ensemble spread. The 1-degree projected decrease for the CCS (-0.96) is larger than the range of the ensemble of projections from the downscaled 12-km model (CCS: -0.86 to -0.94). The 1 degree projected decrease for the N-CCS (-0.76) is smaller or less severe than the range of the ensemble of projections from the downscaled 12-km model (N-CCS: -0.80 to -0.88). The same is true for the surface decline $\Omega_{calcite}$ (Table 1).

At the bottom, Ω decreases in all projections near the coast on the shelves (Table 1; Fig. 5). At the 1.5-km resolution, spatial variability in the magnitude of the projected conditions exists on the shelf. The most severe changes in bottom Ω correspond with regions that experience the largest changes in bottom $O_2$. In the N-CCS, the 1.5-km projected declines are even larger than the 12-km projected declines even when considering the ensemble range. The 1-degree model projected decline in $\Omega_{arag}$ for the CCS (-0.47) is more severe than the downscaled ensemble range (CCS: -0.38 to -0.46) for the downscaled 12-km model.
In the N-CCS, the 1-degree model projected decline in $\Omega_{arag}$ for the N-CCS (-0.32) falls within the range of the ensemble of projections from the downscaled 12-km model (-0.26 to -0.39). In the 1.5-km model projections, the decline is greater than the 1-degree model projects and falls well outside the range of the 12-km projections. This difference between the global and downscaled simulations is even greater on the shelves (indicated as the starred values in Table 1). The same is true for the bottom decline in $\Omega_{calcite}$ (Table 1).






### 3.6 Themes across projected changes for the CCS

All climate-associated stressor variables agree with the 1-degree projections in terms of the direction of the trend, but not the magnitude of the change. The 1-degree model projections for the CCS are largely consistent with the 1 degree model projected global trends with some differences in the nearshore upwelling areas (Fig. S1). All carbon variables are sensitive to the

inclusion of coastal processes which both downscaled projections provide. In addition, all of the projections suggest greater change in most variables in northern regions of the CCS, and in the upwelling regions.

Nitrate increases over much of the domain in the upper 200 m (Fig. 6) in both the high- and medium-resolution downscaled simulations. Nitrate on average increases in the global simulations, but the magnitude and direction varies widely across the

ensemble members. In addition to the projected small increase in nutrient concentrations in the upwelling system of the CCS, the winds are slightly more intense (2 % increase in the magnitude of the wind stress) during the upwelling season (April - September) in the future years. The timing of the onset and duration of the upwelling season in the northern CCS remains the same in the future in these projections. Despite these differences from wind-based upwelling metrics, the in-water upwelling metric CUTI indicates no net change in the upwelling intensity of the future N-CCS in the simulations evaluated here. When

nitrate is included in the upwelling measure, as in BEUTI, there is a slight decline in the upwelling of nitrate (1-2%), commensurate with a decrease in nitrate at the surface in the N-CCS (Fig. 6). This result is sensitive to the distance offshore (20-75km) over which the index is calculated. The direction of the trend does not change, but BEUTI for example, further declines (4%) as bin boundaries move closer to shore. The further offshore the bin extends, the weaker the signal becomes. Both of these measures suggest that the upwelling is not intensified in our projected future, despite the slight increase in winds.

This result is consistent with the results of Howard et al. (in review), where increased stratification in the future simulations impeded increases in upwelling intensity.

The projected temperature change affects the solubility of the gases, generating a solubility-driven decline, and the increased nutrient content would correspond to a stoichiometric oxygen loss as well. Over the entire CCS, the decrease in oxygen over

the upper 200 meters was 0.45 ml/l or 20.10 mmol/m$^3$ (Table 1). The solubility-driven change accounts for most (~67%) of this change (13.42 mmol/m$^3$ using 1.92 change in temperature from Table 1). The additional nitrate brought into the region from the large-scale models (0.79 mmol/m$^3$) corresponds to an additional drawdown of 6.81 mmol/m$^3$ of oxygen. In the N-CCS, the change in oxygen is a bit larger than across the entire CCS – 0.69 ml/l (30.82 mmol/m$^3$, Table 1). The solubility driven changes contribute a bit less (~44%) in the N-CCS than the entire CCS, but the nitrate signal is larger in the N-CCS

corresponding to 11.21 mmol/m$^3$ change in oxygen from an increase of 1.3 mmol/m$^3$ of nitrate in the upper 200 meters of the water column. The result is that the solubility driven changes combined with the increased supply of nutrients to the upper 200 meters of the N-CCS account for 80% of the projected oxygen change in the N-CCS.



Similarly, we would expect carbon content to increase in source waters commensurate with an increase in nutrients and lower
oxygen concentrations. The corresponding stoichiometric increase in DIC to the increase in nutrients (5 mmol/m$^3$) would only
account for a small decline in pH (0.02) or $\Omega$ (0.03). The majority of the pH and $\Omega$ decline is due to anthropogenic carbon
content increase in DIC associated with the RCP 8.5 scenario forcing (95 mmol/m$^3$). Spatial variability in $\Delta$DIC corresponds
with variability in the water buffer capacity and Revelle factor, with southern CCS regions of relatively high buffer capacity
having the greatest rates of DIC uptake (Fig. 8).


TA increases in the future in the subsurface on the shelves of the CCS and even more so on the upper slope (Fig. 8). At the
surface, it declines, and these two changes offset each other in the depth-averaged 200 meter change. These changes in these
depth ranges contribute to the results for the carbon variables in Table 1, impacting different carbon variables differently.
On the shelves of the downscaled simulations, the source waters are further modified by coastal processes including increased
productivity, freshwater delivery and denitrification. The inclusion of these processes causes the bottom waters on the shelf to
experience a more severe increase in pCO$_2$ and declines in oxygen, pH, and $\Omega$ than observed in the shallowest regions of the
global models (starred values in Table 1). The difference between the bottom estimated changes in the CCS in the depth ranges
resolved by the global models and on the downscaled shelves is greatest for the carbon variables.

### 3.7 Modification

In general, the CCS experiences a greater change for most variables at all resolutions than the global ocean. However, only
the carbon variables emerge as amplified or dampened by the downscaled simulations. Across the spread of ensemble members
for the entire CCS in the 12km simulation, the downscaled projected increase for pCO$_2$ (columns C, F, G) is amplified relative
to the 1-degree models (columns B, E) in both the 200 m depth averaged and the surface depth ranges (Table 1). The
downscaled decrease in pH at the surface and the bottom is modified relative to what the 1-degree models project for the CCS
and N-CCS in the downscaled projections. The surface change is less severe than in the 1-degree model, and so is considered
dampened relative to global change. At the bottom, pH in the N-CCS is dampened relative to the 1-degree model. The decrease
in $\Omega$ over the upper 200 meters is less severe than the 1-degree model projection for the N-CCS, and falls outside of the
ensemble range, so is considered dampened relative to global change (Table 1). The N-CCS pCO$_2$ increase is amplified in
both downscaled simulations (12-km, 1.5km), at all depth ranges (Table 1, columns F and G). In both downscaled projections
(12-km, 1.5km), the surface and bottom pH decrease is dampened in the N-CCS relative to the surface pH decline in the 1
degree model for that region (column E). The N-CCS 200 m depth averaged $\Omega$ decrease is dampened in both downscaled
simulations (12-km, 1.5km), and amplified at the surface (Table 1). In the N-CCS region, more of the water column is modified
for the carbon variables.





## 4 Discussion

Globally, under RCP 8.5, the future oceans simulated using CMIP5 are projected on average to be warmer (SST, mean ± 1 SD: 2.73 ± 0.72°C), higher in $pCO_2$, lower in $O_2$ content (RCP8.5: –3.45 ± 0.44%), and more acidified (surface pH –0.33 ± 0.003 units) (Gattuso et al., 2015). Regionally, the CMIP5 models project the North Pacific to be one of the regions to experience the most warming, most severe $O_2$ declines, and largest extents of corrosive conditions (Bopp et al., 2013; Feely et al., 2009; Gattuso et al., 2015; Gruber et al., 2012; Hauri et al., 2013; Long et al., 2016). Here, we present one of the first

downscaled multivariable projections of environmental change out to the end of the century in the CCS driven by a suite of CMIP5 forcings instead of a single global model. While the downscaled projections show changes that are similar in direction to those of the global simulation, the magnitude and spatial variability of the change differs in the coastal process-resolving downscaled projections and to a varying degree depending on the variable, depth range, and subregion of interest.

The CCS experiences a greater change for many variables at all resolutions than the global ocean, however this change is modified in the CCS in both downscaled simulations (12-km, 1.5km), for $pCO_2$, $\Omega_{arag}$ and $\Omega_{calcite}$, and pH. Amplification of global trends within the CCS upwelling systems in the future has been shown before for oxygen specifically (Dussin et al., 2019) and was identified through the response of the downscaled model to a series of idealized experiments with perturbations in the source water oxygen and nutrient concentrations. Source water changes in oxygen drove a two-fold larger change in

oxygen than nutrient supply alone, and both of these drivers were determined to be more important than intensifying winds. In our projections, the more realistic winds were different than in Dussin et al. (2019), with a small intensification of about 2%. The source waters were lower, but the oxygen decrease in the N-CCS fell within the range of the ensemble members explored here. Dussin et al. (2019) only explored one global model (GFDL) as a driver, and as such the definition of amplification differs from the one we use here. Much like experiments conducted by Liu et al. (2012, 2015) for ocean

temperatures and described in Alexander et al. (2020), using a multi-model mean to drive a downscaled ocean model retains the linear component of the climate change forcing only and is not able to assess the range of the response. Fundamentally, our definition of amplification relies on the range of the ocean condition response. The mechanism of remote biogeochemical redistribution and influence via boundary conditions in the CCS remains influential for the carbon variables that were identified as amplified here. While climate stressor variables have been identified as amplified historically using records spanning

several decades or more (Chavez et al., 2017; Osbourne et al., 2020), regional future projections have focused on multi-model mean conditions projected over 100 years into the future.

Although the downscaled model projected a small intensification in the projected upwelling-favorable wind stress of 2% which is consistent with prior work on this topic (Bakun, 1990; Garcia-Reyes et al., 2015; Rykaczewski and Dunne, 2010;

Rykaczewski et al., 2015), no change was quantified in the upwelling fluxes in the water column using the CUTI, and a slight decline was observed in BEUTI (nutrient flux) measures of upwelling. We observe lower oxygen, higher nutrient content in



source waters, but this change does not make it to the surface. Within the CCS, solubility-driven oxygen changes are important, which is consistent with oxygen escape from the ocean being increasingly important in future projections (Li et al. 2020). Moving south within the CCS, solubility increasingly outcompetes nutrient-driven changes in oxygen drawdown. However,

the projections here indicate that the magnitude of oxygen decrease was within the bounds of the ensemble range, and so was not amplified relative to the global models, unlike the carbon variables.

The projected change in pH is consistent with prior pH projections for the CCS downscaled with the same RCP 8.5 scenario using a ROMS model (Gruber et al., 2012; Hauri et al., 2013; Marshall et al., 2017; Turi et al., 2016). These projections were

performed with different biogeochemical models described in Gruber et al. (2012), and Fennel et al. (2006, 2008) and relied on multi-model means or individual ensemble members for the global models. The values we obtained are lower than the projections of Rykaczewski and Dunne (2010) for the CCS. They used GFDL Earth System Model 2.1, a different forcing scenario, and a different biogeochemistry model (Dunne et al., 2007). Despite clear modification of global trends indicated by downscaled projections of $pCO_2$, $\Omega$, and pH, the biogeochemical model formulations differ across resolutions and may be

contributing to the amplified signals. Differences include parameterizations of gas exchange, detritus classes, sinking velocities, as well as benthic boundary conditions; the latter has been identified in prior work to be important for $O_2$ models in coastal regions (Moriarty et al., 2018; Siedlecki et al., 2015). Any of these could contribute to the differences observed across model resolutions. CMIP5 results are based on an ensemble average of many models, which all utilize different formulations and complexities for their ecosystem models, further contributing to the uncertainty provided from the biogeochemical

boundary conditions. Overall, the projected annual, depth-averaged change in pH appears to depend mostly on the anthropogenic forcing scenario, and all the models agree on the direction and relative magnitude despite these differences.

Projected changes in surface and bottom pH, $\Omega$, and $pCO_2$ are modified by the inclusion of coastal processes when downscaling is employed. Coastal processes that influence the variability of carbon variables occur in these regions of the water column

include freshwater delivery and sediment water interactions. At the surface, pH, $\Omega$ and $pCO_2$ change differently. TA declines at the surface, while DIC increases. DIC increases due to increased storage of carbon from the increase in carbon in the future atmosphere. The TA changes are driven in part by the altered timing of the freshet in the N-CCS as well as the presence of a river plume in an upwelling regime. Freshwater in the region is known to be corrosive due to naturally low TA, which impacts the buffering capacity of the surface waters. This result can be seen in the surface difference plots for $pCO_2$ near the Columbia

River plume (Fig. 4) and the surface TA change (Fig. 8). The 12-km projections include climatological freshwater fluxes as precipitation along the coastline instead of resolving river plumes like the 1.5-km projections, but despite these different freshwater parameterizations, both models indicate modification of carbon variables in the N-CCS Cascadia domain with different directions for different variables. While the freshwater amplifies the global rate of change for the surface $pCO_2$ and $\Omega$, over the entire water column (200 m average), the pH change is dampened.




In our regional downscaled simulations, the change in temperature and TA act together to offset the increase in the DIC signal in the coastal upwelling regime (Fig. 8), and for pH these changes offset each other in the upper 200 m of the water column. The global models show very little change in TA in the region. While the downscaled bottom TA change is small (20-50 mmol/m$^3$, Fig. 8), this amounts to an increase in pH of 0.07-0.18 and an increase in $\Omega$ of 0.15-0.21 - enough to offset 40-60%
of the reduction in $\Omega$ due to increased atmospheric $CO_2$ concentrations. At the bottom, the increase in TA modifies the projected pH change in the N-CCS by reducing it.

Both biogeochemical models include denitrification at the sediment water interface which impacts both the nitrogen and TA cycling in the model. Denitrification is a source of TA (Chen et al., 2002) and has been shown to impact shelf wide TA in
other regions (Fennel et al., 2008). In the CCS, denitrification has been observed on the shelf and slope, peaks on the slope, and is greater off the Washington coast than off Mexico (Hartnett and Devol, 2003). As the source waters become lower in oxygen content, denitrification should increase, providing an additional feedback on the nutrients and carbon variables, and dampening the global ocean acidification signal. Bottom $\Omega$ is not amplified in Table 1, but if calculated solely over the shelf region (asterisks in Table 1), then bottom $\Omega$ changes are amplified in both domains. While this source of alkalinity has not
been observed directly in the modern ocean, a source of TA was identified and interpreted as calcium carbonate dissolution in the CCS (Fassbender et al., 2011). These results suggest future projections should consider salinity and TA forcing feedbacks when performing regional projections as these can alter regional impacts of carbon variables.

Different carbon variables are sensitive to different physical climate forcings, a result that is consistent with recent work
suggesting that competing sensitivities may dampen the variability of pH in the future (Jiang et al., 2019; Kwiatkowski and Orr, 2018; Salisbury and Jonsson, 2018; Takahashi et al., 2014). The sensitivities of the various carbon variables to thermal and geochemical (carbon dioxide) forcing was explored in an idealized simulation of future conditions from CMIP5 models in Kwiatkowski and Orr (2018), in the modern ocean in Takahashi et al. (2014) and Jiang et al. (2019), and in the Gulf of Maine by Salisbury and Jonsson (2018). Kwiatkowski and Orr (2018) determined that the balance between the change in DIC
and TA drives the variability of pH, in combination with temperature in mid-to-high latitudes -- effects that largely cancel each other out. While the processes that control TA and DIC are similar, the addition of atmospheric $CO_2$ changes DIC alone, altering the balance between these two reservoirs. In the Gulf of Maine, temperature and salinity anomalies in combination with TA variability offset the long-term OA trend (Salisbury and Jonsson, 2018). Their analysis indicated that $\Omega$ is more sensitive to temperature and salinity variations than pH, and this result was confirmed globally in Jiang et al. (2019).
Kwiatkowski and Orr (2018) also found the seasonal amplitude of $\Omega_{arag}$ is expected to strengthen in some regions and attenuate in others due to the high sensitivity of this variable to temperature.

Spatially, within the CCS, the projections suggest greater change in most variables in northern regions of the CCS, and on the shelves in the annual averages. Currently, the N-CCS is the most productive region of the CCS (Davis et al., 2014; Hickey and



Banas, 2008) and experiences the most prevalent hypoxia (Connolly et al., 2010; Peterson et al., 2013) and corrosive events (Feely et al., 2018; 2016). One aspect that differentiates the region from the rest of the CCS is that the N-CCS experiences both seasonal upwelling and downwelling currently (Hickey and Banas, 2008), which is not projected to change in 2100 under RCP8.5. The continued existence of a downwelling season in the future impacts some variables more than others. The projected change for oxygen and carbon variables is different seasonally. The fall transition and winter mixing re-oxygenates shelf waters

(Siedlecki et al., 2015), and this pattern continues into the future. Hypoxia will continue to exist in the summer months, but for a longer period of the summer. Corrosive, low pH conditions, however, will occur year-round. The fraction of the year during which bottom water on the shelf is corrosive ($\Omega$ <1) or low pH (pH <7.65) nearly doubles. This asymmetry has been identified previously in the Salish Sea (Ianson et al., 2016), and exists because of the difference in equilibration timescales at the surface for oxygen and carbon dioxide.

**5 Conclusions**

We present one of the first multi-model, downscaled multivariable projections of changing temperature, pH, $pCO_2$, $\Omega$, and $O_2$ in the CCS. The downscaled projections are driven by the same forcing as the global simulation, and projected changes for the CCS are consistent with the directional trends indicated by the global model for scenario RCP 8.5 - warmer, more acidified, higher carbon content, and lower oxygen concentration. However, the magnitude and spatial variability of the change differs

in the coastal process-resolving downscaled projections and to a varying degree depending on the variable of interest. Changes in $pCO_2$ concentrations, $\Omega$, and pH are modified in the downscaled projections relative to the projected global simulation, suggesting downscaled projections are necessary to more accurately project future conditions of these variables. The diversity of these projected changes of future ocean conditions emphasizes the need to improve our understanding of mechanisms by which coastal processes interact with these large-scale drivers of change or properly simulate and capture these feedbacks in

projections.

**Code and Data Availability**

Archived model fields will be available from the library at UConn (link) upon acceptance of this publication.

**Author Contribution**

SS, DP, CD formulated the research goals and designed the experiments and analyses. DP and HF carried out the
experiments. TK, SS, JN, and CD acquired the financial support for the project that led to its publication. TK managed and coordinated research meetings with the co-authors. DP and EN performed formal analysis of the model fields. DP created the figures which SS helped design. SS and PM developed the 1.5 km simulations alongside the Coastal Modeling Group at



UW. CD developed the 12 km simulations. RF and SA collected the data used for model evaluation and assisted in determining how model evaluation would be performed. SS prepared the manuscript with contributions and edits from all
co-authors.

## Competing Interests

The authors declare that they have no conflict of interest.

## Acknowledgements

This work was supported by funding from the Schwab Charitable Fund made possible by the generosity of Wendy and Eric
Schmidt, the Washington Ocean Acidification Center, and the following awarded to C.D.: NSF (OCE-1635632, OCE-1847687, OCE-1419323, OCE-1737282), NOAA (NA15NOS47801-86, NA15NOS47801-92,NA18NOS4780167), California Sea Grant and Ocean Protection Council (R/OPCOAH-1), and Gordon and Betty Moore Foundation (GBMF3775). This publication is partially funded by the Joint Institute for the Study of the Atmosphere and Ocean (JISAO) under NOAA Cooperative Agreement NA15OAR4320063, Contribution No. 2018-0183. RAF and SRA were funded by NOAA/PMEL, the
NOAA Ocean Acidification Program, and the former NOAA Global Carbon Cycle Program. We would like to thank Jennifer Sunday for her participation in helpful discussions. The 12km runs were performed on the Cheyenne cluster at NCAR/CISL. The 1.5-km simulations were performed on the University of Washington Hyak supercomputer system, supported in part by the University of Washington eScience Institute. This is PMEL contribution number 4919.

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





| Region | Global | CCS | | | N-CCS | | | |
|---|---|---|---|---|---|---|---|---|
| Model | A. 1 degree | B. 1 degree | C. 12 km | D. 12 km Ensemble spread | E. 1 degree | F. 12 km | G. 1.5 km | H. 12 km Ensemble spread |
| **200m avg** | | | | | | | | |
| $\Delta$Temp (°C) | 1.97 | 1.63 | 1.81, 2.47* | 1.35 to 2.27 | 2.21 | 1.94, 2.18* | 2.32, 2.63* | 1.43 to 2.60 |
| $\Delta$Oxygen (ml/l) | -0.19 | -0.52 | -0.60, -0.57* | -0.50 to -0.70 | -0.56 | -0.69, -0.67* | -0.58, -0.52* | -0.44 to -0.94 |
| $\Delta pCO_2$ ($\mu$atm) | 401 | 492 | **753, 710*** | 682 to 824 | 527 | **908, 920*** | **725, 639*** | 683 to 1044 |
| $\Delta$pH | -0.297 | -0.321 | -0.331, -0.323* | -0.309 to -0.353 | -0.332 | -0.311, -0.326* | -0.323, -0.319* | -0.252 to -0.343 |
| $\Delta\Omega_{Arag}$ | -0.85 | -0.71 | -0.70, -0.69* | -0.65 to -0.75 | -0.62 | *-0.46, -0.53** | *-0.59, -0.63** | -0.37 to -0.51 |
| $\Delta\Omega_{Calcite}$ | -1.33 | -1.11 | -1.10, -1.09* | -1.03 to -1.17 | -0.99 | *-0.73, -0.85** | *-0.93, -1.00** | -0.56 to -0.81 |
| **Surface** | | | | | | | | |
| $\Delta$Temp (°C) | 2.49 | 3.12 | 3.30, 3.24* | 2.53 to 4.07 | 3.15 | 3.29, 3.21* | 2.89, 2.91* | 2.34 to 4.24 |
| $\Delta$Oxygen (ml/l) | -0.23 | -0.32 | -0.35, -0.39* | -0.27 to -0.43 | -0.38 | -0.41, -0.41* | -0.41, -0.40* | -0.29 to -0.53 |
| $\Delta pCO_2$ ($\mu$atm) | 379 | 392 | **434, 434*** | 432 to 436 | 365 | **430, 420*** | **376, 366*** | 425 to 435 |
| $\Delta$pH | -0.309 | -0.319 | *-0.285, -0.292** | -0.284 to -0.286 | -0.343 | *-0.297, -0.297** | *-0.288, -0.284** | -0.294 to -0.300 |
| $\Delta\Omega_{Arag}$ | -0.98 | -0.96 | *-0.90, -0.85** | -0.86 to -0.94 | -0.76 | **-0.84, -0.85*** | **-0.80, -0.74*** | -0.80 to -0.88 |
| $\Delta\Omega_{Calcite}$ | -1.52 | -1.50 | *-1.42, -1.35** | -1.36 to -1.48 | -1.21 | **-1.35, -1.35*** | **-1.28, -1.19*** | -1.30 to -1.40 |
| **Bottom (<500m)** | | | | | | | | |
| $\Delta$Temp (°C) | 1.98 | 1.65 | 1.78, 2.26* | 1.39 to 2.17 | 1.34 | **1.70, 1.86*** | **2.05, 2.34*** | 1.40 to 2.30 |
| $\Delta$Oxygen (ml/l) | -0.22 | -0.43 | -0.56, -0.64* | -0.39 to -0.73 | -0.63 | -0.66, -0.68* | -0.60, -0.61* | -0.37 to -0.96 |
| $\Delta pCO_2$ ($\mu$atm) | 432 | 505 | **783, 883*** | 663 to 903 | 592 | **956, 1029*** | **844, 908*** | 675 to 1164 |
| $\Delta$pH | -0.306 | -0.286 | -0.259, -0.318* | -0.231 to -0.287 | -0.333 | *-0.269, -0.290** | *-0.281, -0.312** | -0.206 to -0.310 |
| $\Delta\Omega_{Arag}$ | -0.68 | -0.47 | *-0.42, -0.59** | -0.38 to -0.46 | -0.32 | -0.35, -0.38* | -0.43, -0.50* | -0.26 to -0.39 |
| $\Delta\Omega_{Calcite}$ | -1.06 | -0.74 | *-0.66, -0.94** | -0.60 to -0.72 | -0.50 | -0.56, -0.61* | -0.69, -0.80* | -0.42 to -0.62 |



**Table 1:** Annual average differences between climate stressor variables in the future and the base/modern conditions for the global ensemble, 12-km, and 1.5-km projections over different regions of the water column (200 m averaged, surface, and bottom < 500 m). Column A depicts the global average from the ensemble average of CMIP5. Column B includes the global (1 degree) ensemble average difference for the CCS region followed by, in column C, the CCS wide difference from the 12km downscaled results. The final CCS column (column D) includes the ensemble spread as the range across the 5-member model spread of the results for the 12-km model projections column for the CCS domain. The N-CCS region results span columns E-H in this table. The next three columns detail the differences in the Cascadia domain for the global ensemble average (column E), the 12-km (column F) and 1.5-km downscaled projections (column G). The final column (column H) includes the ensemble spread as the range across the 5-member model spread of the results for the 12-km model projections column for the Cascadia domain. The direction with which each variable described above was amplified/dampened relative to the global models outside the ensemble variability is highlighted using bold text and dark grey (amplified) and italic text with light grey (dampened) shading. Within the downscaled simulation, values are also provided just on the shelf (< 200 meter isobath) and denoted with an asterisk (*) next to the number.





**Figure 1. Base state climate stressor variables in the CCS in both domains. Depth-averaged over 200 meters values for the climate stressor variables in the base state/modern conditions for the 12-km (CCS-wide, full panel) and 1.5-km simulations (N-CCS, inset) for (a) temperature (deg C) (b) $O_2$ (ml/l) (c) $\Omega_{arag}$ (d) $pCO_2$ (μatm) (e) pH, and (f) $\Omega_{calcite}$.**



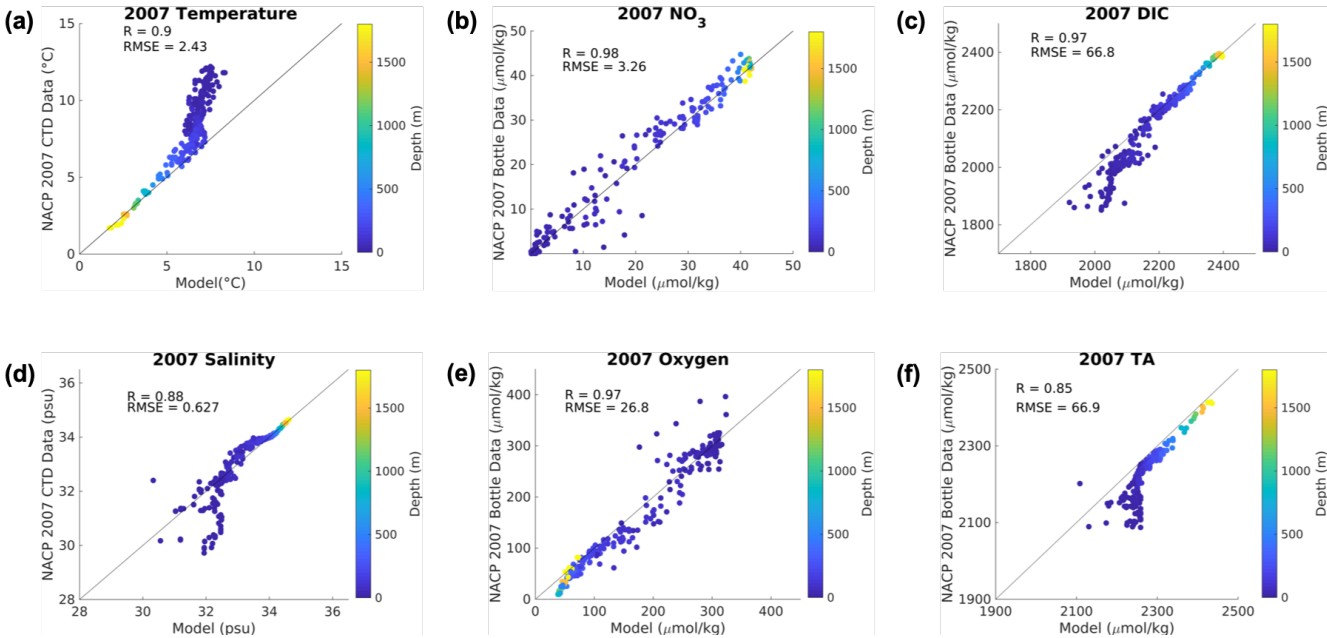

**Figure 2. Model evaluation. Comparison between the bottle data from the 2007 cruise detailed in Feely et al. (2008) on the y-axes, and the simulated parameter (x-axis) from the 1.5-km model forced with the 12-km model at the boundaries. Colors indicate the depth where the bottle sample was sampled. Correlation coefficient (R) and Root Mean Squared Error (RMSE) are reported as well. The units for each variable are labeled on the axes.**



**Figure 3. Projected temperature and oxygen changes in the CCS. Differences between climate stressor variables temperature and**
**oxygen in the future and the base/modern conditions for the 12-km (CCS-wide, full panel) and 1.5-km (N-CCS, inset) projections**
**for (a) SST (deg C) (b) depth-averaged temperature over the upper 200 meters (deg C) (c) bottom temperature (deg C) (d) surface**
**$O_2$ (ml/l) (e) depth-averaged $O_2$ over the upper 200 meters (ml/l) (f) bottom $O_2$ (ml/l). Positive values indicate a change that is greater**
**in the future than the base/present condition and negative values indicate lower values in the future than the base/present conditions**
**depicted in Figure 1.**





**Figure 4. Projected pCO₂ and pH changes in the CCS. Differences between climate stressor variables pCO₂ and pH in the future and the base/modern conditions for the 12-km (CCS-wide, full panel) and 1.5-km (N-CCS, inset) projections for (a) surface pCO₂ (µatm) (b) depth-averaged pCO₂ over the upper 200 meters (µatm) (c) bottom pCO₂ (µatm) (d) surface pH (e) depth-averaged pH over the upper 200 meters and (f) bottom pH. Positive values indicate a change that is greater in the future than the base/present condition and negative values indicate lower values in the future than the base/present condition depicted in Figure 1.**




**Figure 5. Projected Ω changes in the CCS. Differences between climate stressor variables Ω in the future and the base/modern conditions for the 12-km (CCS-wide, full panel) and 1.5-km projections (N-CCS, inset). (a) surface $\Omega_{arag}$ (b) depth-averaged $\Omega_{arag}$ over the upper 200 meters (c) bottom $\Omega_{arag}$ (d) surface $\Omega_{calcite}$ (e) depth-averaged $\Omega_{calcite}$ over the upper 200 meters and (f) bottom $\Omega_{calcite}$. Positive values indicate a change that is greater in the future than the base/present condition and negative values indicate lower values in the future than the base/present condition depicted in Figure 1.**




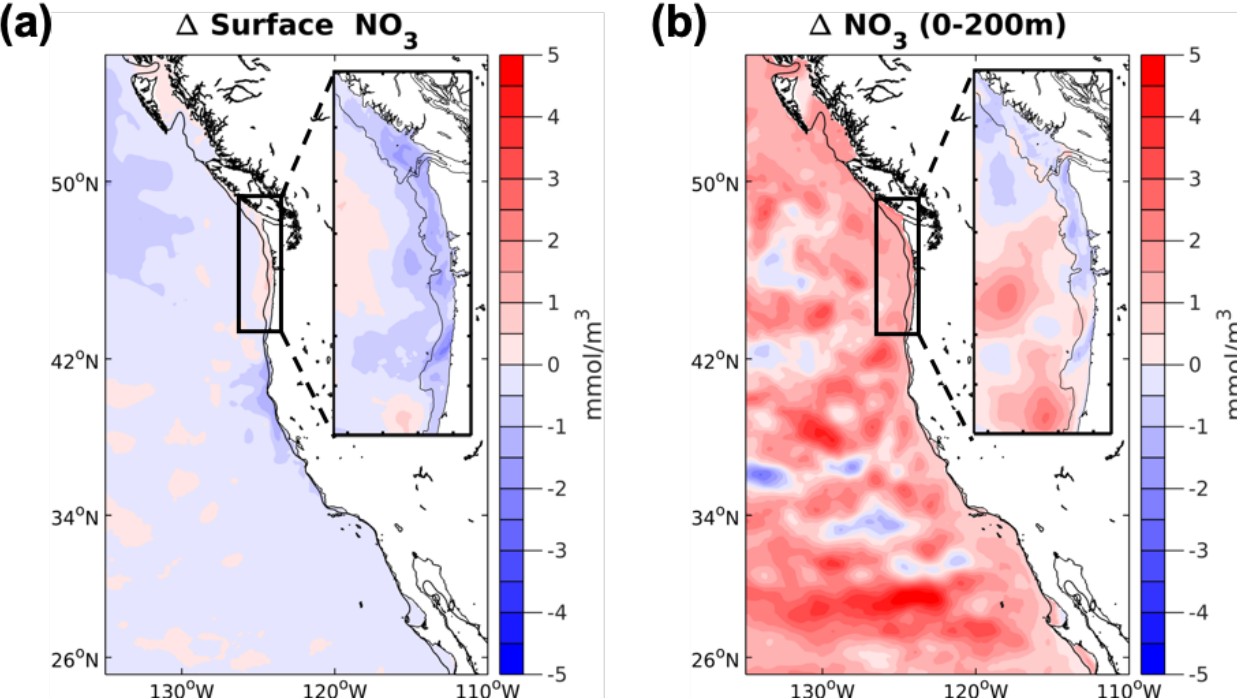

**Figure 6. Projected changes in nitrate for the CCS.** Differences between the future and the base/modern nitrate (NO₃) conditions for the 12-km (CCS-wide, full panel) and 1.5-km (N-CCS, inset) projections at (a) the surface (b) depth-averaged (0-200m). Positive values indicate a change that is greater in the future than the base/present condition and negative values indicate lower values in the future than the base/present condition.



**Figure 7. Projected changes for DIC and TA in the CCS.** Simulated downscaled changes in annual average DIC and TA (mmol/m³)
in the future and the base/modern conditions for the 12-km (CCS-wide, full panel) and 1.5-km (N-CCS, inset) projections. (a)
Surface DIC (b) 200 m depth averaged DIC (c) bottom DIC (d) Surface TA (e) 200 meter depth averaged TA and (f) Bottom TA.
Positive values indicate a change that is greater in the future than the base/present condition and negative values indicate lower
values in the future than the base/present condition.
