# Peer review of "Coastal processes modify projections of some climate-driven stressors in the California Current System"

_Biogeosciences, 2020_

## Referee Comment (RC1) · Anonymous Referee #1 · 30 Aug 2020

GENERAL COMMENTS ON MANUSCRIPT bg-2020-279

This multi-model, downscaled projection of changes in the carbonate system by Siedlecki et al. represents a substantial and timely contribution to the biogeosciences community. It fits nicely within the scope of the journal and should be of interest to a large group of readers. The authors put their own work in the context of the previous literature and they use a sound methodology. The structure of the manuscript is intuitive, and the results are appropriately presented and discussed (for the most part; see below).

The "multi-model" aspect of the manuscript is both a curse and a blessing. While

[Figure]

we learn quite a bit from this intercomparison, there are multiple instances where the manuscript lacks crucial clarifications. I make several suggestions below on how to clarify the parts where I stumbled, and I believe that the manuscript will be substantially improved if the authors address those points.

I also note that the manuscript submission was rushed. Sections 3.6,3.7,4 repeatedly refer to a "Fig.8" that doesn't exist in the manuscript PDF. Line 85 refers to a "model evaluation provided as supplemental material" but there are no model evaluation in the Supplement. Line 195 refers to colors in Table 1 while Table 1 is in grayscale. Considering that there are 11 authors on the manuscript, I don't understand how none of them was willing to re-read the manuscript before submitting it? Reviewers are supposed to review the science, not proofread.

SPECIFIC COMMENTS: MAJOR

(1) The comparison between the projections of the 1-degree, 12km, and 1.5km models is a major focus of the manuscript. However, I'm still unclear as to what is compared to what. If I interpret lines 178-184 correctly, each model (1-degree, 12km, 1.5km) is considering a different time period:

1-degree: 1971-2000 (present) versus 2071-2100 (future) 12km: 1994-2007 (present) versus 2085-2100 (future) 1.5km: 2002-2004 (present) versus 2094-2096 (future)

If I'm correct, then the differences in time periods contribute to the differences in model projections. I don't consider this mismatch in time periods a deal breaker, but I certainly think that it should be emphasized and discussed up front in the manuscript?

To add to the confusion, line 185 says: "Comparisons between the 12 and 1.5km-resolution simulations were made with the same year span". What "comparisons" are we talking about? Are we referring to the present state shown in Figure 1? Or are we referring to all figures that show the 12km and the 1.5km together? (Figures 1-7?) What about Table 1 (which includes a "comparison" between the 12km and the 1.5km)?

Again, this sort of comparison is supposed to be a major focus of the manuscript and thus these points must be clarified. I would recommend that the time span used for each model configuration be clearly stated in the caption of Table 1 and Figure 1-7.

(2) Figure 8, referenced multiple times in the text, doesn't exist in the PDF document.

(3) Line 146: "The 12-km historical simulation forcing is described in Renault et al (in review)..."

This reference does not exist in the "References" section and therefore is not available to the reviewers.

(4) Lines 155-156: "Initial and boundary conditions had the same kind of centennial trend addition for temperature, salinity, and all biogeochemical tracers (O2, nitrate, phosphate, silica, iron, dissolved inorganic carbon, alkalinity)."

This statement is vague and leaves much to interpretation. What about adding one little table in the "Supplement" that details what "centenial trend" was assumed for each of those variables? Were the trends assumed constant in time (linear trend)? Constant in space?

(5) In figure 3a,b,c, the authors are comparing temperature changes across 3 depths (surface, 0-200m, bottom) using 3 different colorscales. The fact that they use 3 different colorscales for the same variable (temperature) makes it unecessarily difficult to compare Figure 3a,b,c. The same problem arises in Figures 4 and 5 (pCO2, pH, Omega). It should be possible to find a compromise, meaning find a colorscale that works reasonably well for the three depths. If you don't make such a modification, it becomes unecessarily hard for the reader to get a sense of how the changes vary along the vertical dimension.

SPECIFIC COMMENTS: MINOR

(6) The text of the abstract uses the symbol Omega without defining what it represents.

(7) Lines 48-50: "Warming impacts O2 in other ways, for example by raising organismal metabolic rates and accelerating O2 consumption, and by increasing water column stratification and thus reducing mixing and ventilation"

Is temperature really playing a dominant role in water column stratification in this coastal system? Aren't river inputs and salinity playing a more important role?

(8) Lines 81-83: "We produce multi-model regionally downscaled climate projections of multiple climate-associated stressors (temperature, O2, pH, Omega, and CO2) that resolve coastal processes to create 100-year projections..."

If I'm not mistaken, each model configuration (1-degree, 12km, 1.5km) is considering a different time span (lines 178-184), and only the 1-degree configuration corresponds exactly to a 100-year projection (it's more like a 90-year projection in the case of the 12km and the 1.5km models). I think you should include a $\sim$ symbol in front the "100-year projection" to acknowledge these differences between the 3 model configurations. The same comment applies to Line 179.

(9) Line 85: "The model evaluation, provided as supplemental material..." I don't see a model evaluation in the Supplemental material. Please delete this passage, or add a model evaluation to the Supplemental material.

(10) Lines 101-103: On line 103, please add a note such as: "The CMIP models are further described in Section 2.3."

(11) Line 126: Typo (...between mode resolutions...)

(12) Lines 147-148: "The 12-km projection was forced by adding a monthly climatological difference between CMIP5 RCP 8.5 scenario forcing and the historical run forcing, averaged over 2071-2100 and 1971-2000, respectively"

Doesn't this correspond to the "Delta approach" (or "Delta method")? Wouldn't it be worth mentioning it since it is a common approach for downscaling?

[Figure]

(13) Line 170: "For the future conditions, atmospheric CO2 concentration (800 ppm), and future atmosphere..."

Where is the 800ppm coming from? Is it the difference between 1971-2000 and 2071-2100 in the median of the CMIP models? Please clarify this statement.

(14) Lines 194-195: "is highlighted using pink (amplified) and blue (dampened) colors in Table 1"

Please update the sentence (Table 1 is in grayscale).

(15) Line 202: "1.2.1 Subsubsection (as Heading 3)"

Please delete.

(16) Lines 206-207: "The largest temperature increase, nearly 3 degrees C, occurs on the shelf in both regions of the projections (Fig. 3"

I think the sentence (as currently formulated) is misleading. The "shelf" is defined by bottom depths of *less* than 200m (lines 132-133). So the offshelf region in Figure 3b is representative of the depth interval 0-200m, while the "onshelf" region in Figure 3b could be representing something like 0-100m. Given a strongly stratified variable like temperature, the mismatch in depth can explain the constrast between the off-shelf/onshelf regions in Figure 3b.

(17) Line 252: "All projections show an onshore-offshore trend in pCO2..."

Since you are discussing spatial differences, I would suggest replacing the word "trend" by "gradient".

(18) Lines 334-336: "When nitrate is included in the upwelling measure, as in BEUTI, there is a slight decline in the upwelling of nitrate (1-2%), commensurate with a decrease in nitrate at the surface in the N-CCS (Fig. 6)"

Isn't this a circular reasoning? There is a decrease in surface nitrate (Fig.6), and when

we take into account this decrease in our upwelling metric, we get a "slight decline in the upwelling of nitrate"?

(19) Lines 364-365: "On the shelves of the downscaled simulations, the source waters are further modified by coastal processes including increased productivity, freshwater delivery and denitrification."

I don't question the statement, but these are things that were not shown/demonstrated in the manuscript. Please add "(not shown)" at the end of the statement.

(20) Line 369: "3.7 Modification"

Please replace this heading by something less cryptic, e.g., "Differences between the global and downscaled projections".

(21) Line 401: "In our projections, the more realistic winds were different than in Dussin et al..."

Please clarify this statement. I don't understand what "more realistic winds" you are referring to.

(22) Lines 147-148 were describing a "Delta approach" where the 12km and 1.5km models use the same winds in "present" and "future" times, except for the addition of a "Delta" computed from a 1-degree resolution global model.

Now Line 497 suggests something completely different—that the 12km and 1.5km models were directly using the winds from the 1-degree resolution global model: "The downscaled projections are driven by the same forcing as the global simulation" Which one is right? This must be clarified.

(23) Figure 4a,b,c: Would it be possible to add a little "tick mark" on the colorscale, indicating the projected change in atmospheric $CO_2$ concentration between the "present" (1971-2000) and "future" (2071-2100) periods (according to the median of the CMIP models)? This would provide some perspective on the magnitude of the changes in

pCO2 shown in Figure 4a,b,c.

---

## Referee Comment (RC2) · Anonymous Referee #2 · 12 Oct 2020

I think that this manuscript is an extremely useful contribution to the field, using a collection of projections at different spatial scales to identify how future biogeochemical change will be impacted by coastal processes operating at finer spatial scales. The analysis also benefits by its inclusion of a range of future climate projections, which allows for a quantification of the differences between models that is in excess of variations within the future climate projections beyond that of a difference from a mean future state. Below I make some general comments on the level of detail that the analysis includes – especially with regard to a more detailed quantification of why the local changes occurred and the specific effect of the coastal processes. I also make some specific comments on the discussion.

[Figure]

General comments: (1) I think that this paper is a little more descriptive than I was expecting, or maybe hoping for. The aim of the paper is to describe how models that include higher spatial resolution of "coastal processes" lead to amplified or dampened responses to future change in comparison to coarser-scale models. The paper successfully describes how a model comparison can achieve this useful goal, but I think it falls a little short of clearly and specifically describing what processes and mechanisms lead to the amplified or dampened changes, based on the simulation results.

Let me try to give some specific examples: (a) Starting on line 442, there is a discussion of the differences in freshwater inputs and the related TA effects that reads "The TA changes are driven in part by the altered timing of the freshet in the N-CCS as well as the presence of a river plume in an upwelling regime. Freshwater in the region is known to be corrosive due to naturally low TA, which impacts the buffering capacity of the surface waters. This result can be seen in the surface difference plots for pCO2 near the Columbia River plume (Fig. 4) and the surface TA change (Fig. 8). The 12-km projections include climatological freshwater fluxes as precipitation along the coastline instead of resolving river plumes like the 1.5-km projections, but despite these different freshwater parameterizations, both models indicate modification of carbon variables in the N-CCS Cascadia domain with different directions for different variables." What exactly is the reader to take away from this? How is TA actually changing on the N-CCS (up or down), how does the altered representation of the freshwater effect TA and why is it important that a river discharges into an upwelling region? These questions are not addressed, and the paragraph ends with "both models indicate modification of carbon variables in the N-CCS Cascadia domain with different directions for different variables." It is not clear what "different directions for different variables" actually refers to – pH? TA? One goes up and the other down? And how does the freshwater control this specifically? The text in the next paragraph beginning on Line 451 gives a specific example, so perhaps this example can be clearly wrapped into the paragraph before it starting on line 442, especially if that paragraph is more explicit about the nature of the climatological freshwater input versus the river plume, what specific effect on TA this

change brings, perhaps describing what exactly the results in Fig 4 and 8 illustrate to support this conclusion. (b) Starting on line 485, downwelling is raised as a factor in modulating the response of the N-CCS to future change, especially the fact that it may not change in the future. But it is not clear how the specific details that are subsequently mentioned, namely winter mixing, higher hypoxia, and seasonally persistent corrosiveness, relate to downwelling (or upwelling). Perhaps this might be obvious to a reader with detailed knowledge of the region, or that one is expected to assume how upwelling or downwelling might modulate the coastal response to climate change, but I think it needs to be more clearly organized. Furthermore, the paragraph begins with stating how the shelves and the N-CCS are projected to have greater change, but I don't think the association of these greater changes with upwelling/downwelling is clearly articulated in the paragraph. I also don't see how a specific mechanism is quantitatively related to the greater change based upon the results of the simulations the authors ran, and what the processes were in the better-resolved models that represented the process well enough to generate these changes.

(2) There are many places in the manuscript where statements like this are made: Line 362: "These changes in these depth ranges contribute to the results for the carbon variables in Table 1, impacting different carbon variables differently." These statements are too vague to be helpful, and in the case of this specific sentence, I expected the authors to then elaborate on what variables were different, how they were different, and where they were different, but that is not really achieved in the following sentences. This may sound picky, but I encourage the authors to examine these types of statements and see how they can make them more specific, more informative, and more quantitative.

(3) I would like to see the authors try and articulate some clear and specific conclusions of the paper. I understand their main point that resolving coastal processes matters for future projections, but there are places where the specifics of how they "matter" for the CCS for a given variable in the future could be more clearly stated. For example, in the concluding paragraph, it is written "Changes in pCO2 concentrations, $\Omega$, and pH

are modified in the downscaled projections relative to the projected global simulation, suggesting downscaled projections are necessary to more accurately project future conditions of these variables." So, how are they modified? How do you think carbonate chemistry will be different in the CCS now that you have more resolved models, in contrast to what the global models say? More OA? Less OA? More seasonally variable OA?

Specific comments:

Line 22: The abstract sentence that begins with "These processes. . ." is a little confusing. Are the waters just generally low in oxygen and nutrients, or are the oxygen and nutrient concentrations in those waters projected to change, and thus work in concert with solubility change to alter future conditions? I think a small edit to the sentence will help clarify this.

Line 24: "coastal process resolving projections" is a mouthful and unclear, (and I understand word limits), but how about "projections that resolve coastal processes"?

Line 41-42: So is the SST decline from Lima and Wethey 2012 predicted from a model? Or from a global-scale analysis that may not include local observations? It is not clear why this would differ from the Chavez record. Please add a sentence that describes why these two records give contrasting results.

Line 83-87: This text seems out of place here, and perhaps should be moved to the section of the methods where you describe the three models.

Line 125: What do you specifically mean by "spatially-weighted" means? Why calculate them?

Line 126: "mode" should I think be "model"

Line 155: It is not clear that it is clear what "centennial trend addition" means here. Are there clear trends in these boundaries? Where do they come from?

Line 163: the text ". . ..,2007 with a one year. . .." just reads awkwardly and is confusing. Was the year 2007 run for a year and then compared to observations from 2007?

Line 168: You make the argument, perhaps fairly, that any bias in the simulation generated by the configurations here will be the same in the two periods you compare. It would be more convincing, and help the reader if the reason for this bias was identified. Can this be evaluated? Is it simply a bias in the forcing?

Line 186: Can you add a brief rationale for why you did not, for consistency sake, use the same 3-year future timeframe to compare all of the three models (used 30 years for coarse-scale model)? It seems unnecessary to add in this potential bias, but if bias is not an issue or there is another rationale for using all 30 years of the coarse scale run, please describe.

Line 197: I think it would help to state the stressor variables parenthetically in this sentence.

Line 346: Perhaps there is a convention in the language of this upwelling system that I am unfamiliar with, but why is higher NO3 associated with more O2 drawdown? Is it because the NO3 increase is a tracer of upwelling that can be linked to O2 source water that has a certain, lower O2 signature? Or is this NO3 assumed to be taken up by phytoplankton growth and subsequently used to drawdown O2 at depth?

Line 374: The word "modified" is used to describe the relative pH changes, but the text that follows seems to consistently describe dampening. Can't you replace "modified" with "dampening" to be clearer?

Line 426: It is unclear if "values" refers to the delta pH or the mean pH when comparing to the other studies.

Line 428: Please clarify that you are describing your "downscaled projections" here, and not those of Dunne

Line 458: Here is a place where some specific model details might provide quantitative

information to support the discussion. Denitrification is raised as a process that can affect TA, but the actual differences in denitrification (and its TA effect) in the model simulations are not shown. I understand that there is a limited amount of information to be shown in any paper, but this discussion would be more compelling if the potential denitrification change was reported. Maybe it is a weak effect, maybe strong, and it would be helpful to know.

Figure 4: Have you considered plotting these deltas on a percentage scale? I understand why they are plotted the way they are, to show absolute changes, but the scales are different (necessarily?) across the depths and this makes it a little harder to compare them, if one wanted.

---

## Author Comment (AC1) · 24 Nov 2020

We thank both reviewers for their insightful comments, which helped to improve the manuscript. Please find our point by point responses in the following proceeded by the word "RESPONSE". Please note that all of the figures and values in the tables were updated as the runs needed to be redone. One of the ensemble members required an update and so all the downscaled simulations including those forced by the ensemble mean were updated. The patterns and results did not change as a result of this despite the numbers changing.

Reviewer 1: GENERAL COMMENTS ON MANUSCRIPT bg-2020-279 This multi-

model, downscaled projection of changes in the carbonate system by Siedlecki et al. represents a substantial and timely contribution to the biogeosciences community. It fits nicely within the scope of the journal and should be of interest to a large group of readers. The authors put their own work in the context of the previous literature and they use a sound methodology. The structure of the manuscript is intuitive, and the results are appropriately presented and discussed (for the most part; see below). The "multi-model" aspect of the manuscript is both a curse and a blessing. While we learn quite a bit from this intercomparison, there are multiple instances where the manuscript lacks crucial clarifications. I make several suggestions below on how to clarify the parts where I stumbled, and I believe that the manuscript will be substantially improved if the authors address those points.

I also note that the manuscript submission was rushed. Sections 3.6,3.7,4 repeatedly refer to a "Fig.8" that doesn't exist in the manuscript PDF. Line 85 refers to a "model evaluation provided as supplemental material" but there are no model evaluation in the Supplement. Line 195 refers to colors in Table 1 while Table 1 is in grayscale. Considering that there are 11 authors on the manuscript, I don't understand how none of them was willing to re-read the manuscript before submitting it? Reviewers are supposed to review the science, not proofread.

RESPONSE: We apologize for the confusion this caused the reviewer and appreciate you taking the time to communicate this to us. There was some reshuffling of figures that happened last minute. Specifically, a figure was moved to the supplement and a supplementary figure was moved into the paper. In addition, the journal (Biogeosciences) had requested we alter the table to be in greyscale after we had submitted the manuscript, so the text was not corrected. We have rectified the situation now. Figure 8 was meant to be Figure 7, which is included. References to Figure 8 have been eliminated and Figure 7 references added in their place. The text was updated to reflect that the table is now greyscale, and the model evaluation in supplement was altered to indicate that it is included in Figure 2. Thank you again for giving us the
opportunity to clarify and improve the paper as a result.

SPECIFIC COMMENTS: MAJOR (1) The comparison between the projections of the 1-degree, 12km, and 1.5km models is a major focus of the manuscript. However, I'm still unclear as to what is compared to what. If I interpret lines 178-184 correctly, each model (1-degree, 12km, 1.5km) is considering a different time period: 1-degree: 1971-2000 (present) versus 2071-2100 (future) 12km: 1994-2007 (present) versus 2085-2100 (future) 1.5km: 2002-2004 (present) versus 2094-2096 (future) If I'm correct, then the differences in time periods contribute to the differences in model projections. I don't consider this mismatch in time periods a deal breaker, but I certainly think that it should be emphasized and discussed up front in the manuscript?

RESPONSE:The reviewer is correct in that the time periods do not match exactly. We have included a table below to clarify and added this information to Table 1 as suggested. The difference between the 12km and 1.5km simulations in time is smaller than the reviewer suggests, and so we have clarified this in the methods text in addition to adding it to Table 1.

While these additions do emphasize these differences a bit more, as the reviewer suggests, the differences in time do not account for the modification we find for the carbon variables. The difference between the carbon dioxide in the present (26 ppm) and future (112-136 ppm) between the 1 degree model and the 12 or 1.5 km simulations is not large enough to account for the modification we see in the runs. $pCO_2$ from the 12km simulations in the upper 200 meters, for example, is 261 ppm greater than the 1 degree simulations project for the entire CCS. The modifications highlighted in Table 1 far exceed this difference in atmospheric carbon dioxide.

Model Time period (present, future) CO2 range (present, future) 1-degree 1971-2000 (present) versus 2071-2100 326-369 (present) ; 685-936 (future).

12 km [2002-2004 (present) versus 2094-2096 (future)] 372-377 (present); 802 (future). 1.5 km 2002-2004 (present) versus 2094-2096 (future) 371 (present); 800 (future)

To add to the confusion, line 185 says: "Comparisons between the 12 and 1.5kmres-olution simulations were made with the same year span". What "comparisons" are we talking about? Are we referring to the present state shown in Figure 1? Or are we referring to all figures that show the 12km and the 1.5km together? (Figures 1-7?) What about Table 1 (which includes a "comparison" between the 12km and the 1.5km)? Again, this sort of comparison is supposed to be a major focus of the manuscript and thus these points must be clarified. I would recommend that the time span used for each model configuration be clearly stated in the caption of Table 1 and Figure 1-7.

RESPONSE: We appreciate the opportunity to clarify. The text was updated to reflect that the comparisons are between the downscaled models and the global models, but that the time frames on the downscaled simulations used the same time interval when this was done "Comparisons between the 12- and 1.5-km-resolution simulations and the 1 degree models were made using the same year span despite runs existing for a broader range of years for the 12-km simulation." Table 1 and Figure captions were also updated with time spans as suggested.

(2) Figure 8, referenced multiple times in the text, doesn't exist in the PDF document.

RESPONSE: We again apologize for the confusion this caused the reviewer and appreciate you taking the time to communicate this to us. Figure 8 was meant to be Figure 7, which is included. References to Figure 8 have been eliminated and Figure 7 references added in their place.

(3) Line 146: "The 12-km historical simulation forcing is described in Renault et al (in review)..." This reference does not exist in the "References" section and therefore is not available to the reviewers.

RESPONSE: We again apologize for this oversight and appreciate you taking the time to communicate this to us. We have added this reference to the list.

(4) Lines 155-156: "Initial and boundary conditions had the same kind of centennial trend addition for temperature, salinity, and all biogeochemical tracers (O2, nitrate, phosphate, silica, iron, dissolved inorganic carbon, alkalinity)." This statement is vague and leaves much to interpretation. What about adding one little table in the "Supplement" that details what "centenial trend" was assumed for each of those variables? Were the trends assumed constant in time (linear trend)? Constant in space?

RESPONSE: Boundary conditions came from an ensemble of CMIP5 members, each with their own trend. The centennial trend in each was referred to here. A reference was added to the manuscript where more information on this trend can be found (Howard et al. 2020, Table 1).

(5) In figure 3a,b,c, the authors are comparing temperature changes across 3 depths (surface, 0-200m, bottom) using 3 different colorscales. The fact that they use 3 different colorscales for the same variable (temperature) makes it unecessarily difficult to compare Figure 3a,b,c. The same problem arises in Figures 4 and 5 (pCO2, pH, Omega). It should be possible to find a compromise, meaning find a colorscale that works reasonably well for the three depths. If you don't make such a modification, it becomes unecessarily hard for the reader to get a sense of how the changes vary along the vertical dimension.

RESPONSE:Excellent idea. We have implemented this suggested change and have uploaded new figures for 3,4,5 with an updated color scale. Thank you again for helping to make the manuscript clearer.

SPECIFIC COMMENTS: MINOR (6) The text of the abstract uses the symbol Omega without defining what it represents.

RESPONSE: The text was updated with "saturation state" prior to the first use of $\Omega$.

(7) Lines 48-50: "Warming impacts O2 in other ways, for example by raising organismal metabolic rates and accelerating O2 consumption, and by increasing water column

stratification and thus reducing mixing and ventilation" Is temperature really playing a dominant role in water column stratification in this coastal system? Aren't river inputs and salinity playing a more important role?

RESPONSE:It depends on the season. During the summer months, temperature gradients generated by upwelling intensity largely drive the stratification, while in the winter/spring months when the discharge is highest in the N-CCS, salinity contributes to the stratification structure more. In these simulations, the total discharge was unaltered in the future, only the timing was changed to represent an earlier freshet. The resulting dynamics made warming very important.

(8) Lines 81-83: "We produce multi-model regionally downscaled climate projections of multiple climate-associated stressors (temperature, O2, pH, Omega, and CO2) that resolve coastal processes to create 100-year projections..." If I'm not mistaken, each model configuration (1-degree, 12km, 1.5km) is considering a different time span (lines 178-184), and only the 1-degree configuration corresponds exactly to a 100-year projection (it's more like a 90-year projection in the case of the 12km and the 1.5km models). I think you should include a âĹij symbol in front the "100- year projection" to acknowledge these differences between the 3 model configurations. The same comment applies to Line 179.

RESPONSE: The $\sim$ symbol was added to clarify the approximate time difference. Thank you again for this suggestion.

(9) Line 85: "The model evaluation, provided as supplemental material..." I don't see a model evaluation in the Supplemental material. Please delete this passage, or add a model evaluation to the Supplemental material.

RESPONSE: We apologize for the confusion this caused the reviewer and appreciate you taking the time to communicate this to us. There was some reshuffling of figures that happened last minute. Specifically, a figure was moved to the supplement and a supplementary figure was moved into the paper. The text has been updated to reflect

this change and the current location of the figures in the manuscript.

(10) Lines 101-103: On line 103, please add a note such as: "The CMIP models are further described in Section 2.3."

RESPONSE: We have implemented this suggestion.

(11) Line 126: Typo (...between mode resolutions...)

RESPONSE: We have corrected this typo, and thank the reviewer again for their vigilance.

(12) Lines 147-148: "The 12-km projection was forced by adding a monthly climatological difference between CMIP5 RCP 8.5 scenario forcing and the historical run forcing, averaged over 2071-2100 and 1971-2000, respectively" Doesn't this correspond to the "Delta approach" (or "Delta method")? Wouldn't it be worth mentioning it since it is a common approach for downscaling?

RESPONSE: Yes. We have edited the methods to reflect this change and referenced Alexander et al. 2020 for the method. Alexander, M. A., S. Shin, J. D. Scott, E. Curchitser, and C. Stock, 2020: The Response of the Northwest Atlantic Ocean to Climate Change. J. Climate, 33, 405–428, https://doi.org/10.1175/JCLI-D-19-0117.1.

(13) Line 170: "For the future conditions, atmospheric CO2 concentration (800 ppm), and future atmosphere..." Where is the 800ppm coming from? Is it the difference between 1971-2000 and 2071- 2100 in the median of the CMIP models? Please clarify this statement.

RESPONSE: That's approximately the value in 2085 under RCP 8.5 (range from the time period for the CMIP 5 models =685-936) for future.

(14) Lines 194-195: "is highlighted using pink (amplified) and blue (dampened) colors in Table 1" Please update the sentence (Table 1 is in grayscale).

RESPONSE: We apologize for the confusion. After we submitted the table in color,

the journal (Biogeosciences) had requested we alter the table to be in greyscale, but the already submitted text was not corrected. We have rectified the situation now and appreciate the reviewer's suggestion to do so.

(15) Line 202: "1.2.1 Subsubsection (as Heading 3)" Please delete.

RESPONSE: We have implemented this suggestion.

(16) Lines 206-207: "The largest temperature increase, nearly 3 degrees C, occurs on the shelf in both regions of the projections (Fig. 3" I think the sentence (as currently formulated) is misleading. The "shelf" is defined by bottom depths of *less* than 200m (lines 132-133). So the offshelf region in Figure 3b is representative of the depth interval 0-200m, while the "onshelf" region in Figure 3b could be representing something like 0-100m. Given a strongly stratified variable like temperature, the mismatch in depth can explain the constrast between the offshelf/onshelf regions in Figure 3b.

RESPONSE: We have removed this sentence entirely.

(17) Line 252: "All projections show an onshore-offshore trend in pCO2..." Since you are discussing spatial differences, I would suggest replacing the word "trend" by "gradient".

RESPONSE: We have implemented this suggestion.

(18) Lines 334-336: "When nitrate is included in the upwelling measure, as in BEUTI, there is a slight decline in the upwelling of nitrate (1-2%), commensurate with a decrease in nitrate at the surface in the N-CCS (Fig. 6)" Isn't this a circular reasoning? There is a decrease in surface nitrate (Fig.6), and when we take into account this decrease in our upwelling metric, we get a "slight decline in the upwelling of nitrate"?

RESPONSE: Notably, this sentence does not try to distinguish or attribute the cause of the decrease nitrate at the surface, but instead merely points out that the decrease in nitrate at the surface is consistent with a decline in the upwelling metric which includes nutrients (BEUTI). To further clarify this point we have altered the language of the

sentence to the following: " When nitrate is included in the upwelling measure, as in BEUTI, there is a slight decline in the upwelling of nitrate (1-2%), consistent with a decrease in nitrate at the surface in the N-CCS (Fig. 6)"

(19) Lines 364-365: "On the shelves of the downscaled simulations, the source waters are further modified by coastal processes including increased productivity, freshwater delivery and denitrification." I don't question the statement, but these are things that were not shown/demonstrated in the manuscript. Please add "(not shown)" at the end of the statement.

RESPONSE: We have implemented this suggestion

(20) Line 369: "3.7 Modification" Please replace this heading by something less cryptic, e.g., "Differences between the global and downscaled projections".

RESPONSE: We have implemented this suggestion but edited it slightly to read "Differences between the global and downscaled projections: the impact of including coastal processes in regional projections."

(21) Line 401: "In our projections, the more realistic winds were different than in Dussin et al..." Please clarify this statement. I don't understand what "more realistic winds" you are referring to.

RESPONSE: Our wind fields included monthly wind anomalies derived from the CMIP5 model outputs added to hourly wind fields from the Weather Research and Forecasting Model (the ROMS hindcast forcing). This leads to variability at multiple timescales (including the hourly to daily timescales important for gas exchange fluxes of biogeochemically relevant variables) combined with climate-driven shifts informed directly by the ESM outputs.

This is a "realistic" approach to the changing winds, as opposed to the approach in Dussin et al. 2019: "add 10% to the meridional wind, over the whole domain, only when blowing southward from early June to late September....The magnitude of the perturbation is chosen to be consistent with models projecting increases in upwelling strength under climate change." However, climate models project a more complicated (and compensating) picture of wind-driven changes: e.g. in this and the related manuscript (Howard et al. 2020), across several CMIP5 models, large wind-driven increases in upwelling favorable winds are only found in the springtime northern and central CCS. But these increases are compensated by increased winter downwelling favorable winds (in the northern CCS), and summertime decreases in upwelling in the central CCS (where wind changes oppose upwelling, rather than strengthen it as assumed above).

In this context, the Dussin et al. 2019 approach to winds is less realistic, though still a valuable idealized model experiment. Indeed, the two approaches broadly led to similar model outcomes, which helps to reinforce the conclusions of Dussin et al. 2019 about the importance of remote biogeochemical forcing (and, as a corollary, the lower sensitivity to wind-driven changes).

(22) Lines 147-148 were describing a "Delta approach" where the 12km and 1.5km models use the same winds in "present" and "future" times, except for the addition of a "Delta" computed from a 1-degree resolution global model. Now Line 497 suggests something completely different—that the 12km and 1.5km models were directly using the winds from the 1-degree resolution global model: "The downscaled projections are driven by the same forcing as the global simulation" Which one is right? This must be clarified.

RESPONSE:The description of the winds in response to comment 21 (CMIP5 anomalies added to WRF hindcast winds) may help address this comment; the first description is correct, though another detail is that the wind-current coupling is parameterized rather than run as a fully coupled ocean-atmosphere (Renault et al. 2020 and references therein). The Deltas (CMIP5 2100-2000 changes) are interpolated to the higher resolution WRF grid, and added onto the hourly WRF wind fields. This description was added to the methods. "Atmospheric conditions including air-temperature at the sea surface, precipitation, and downwelling radiation are derived from an uncoupled

Weather Research Forecast model output (c3.6.1; Skamarock et al. 2008) as in Renault, Hall, & McWilliams (2016) and Renault et al. (2020) with more information in Howard et al (in review). To avoid the computational cost of a fully-coupled ocean-atmosphere model, wind and mesoscale current feedbacks are parameterized with a linear function of the surface wind stress as in Renault, Molemaker, et al. (2016). This linear relationship is supported by observations in the CCS (Renault et al. 2017).". The CMIP anomalies are taken at as high a resolution as possible, and are daily. The second statement at Line 497 commented on above was corrected to reflect this clarification "The downscaled projections are driven by the same delta forcing as the global simulation".

Just to clarify, the wind is not the dominant driver for many of the changes presented in this manuscript. So while the methods will be clarified as requested, the big-picture answers are insensitive to these details.

(23) Figure 4a,b,c: Would it be possible to add a little "tick mark" on the colorscale, indicating the projected change in atmospheric $CO_2$ concentration between the "present" (1971-2000) and "future" (2071-2100) periods (according to the median of the CMIP models)? This would provide some perspective on the magnitude of the changes in paper $pCO_2$ shown in Figure 4a,b,c.

RESPONSE: Yes, we have implemented this suggestion.

| Region | Global | CCS | | | N-CCS | | | |
|---|---|---|---|---|---|---|---|---|
| Model [Time Interval]; atmospheric $CO_2$ (present); atmospheric $CO_2$ (future); | A. 1 degree [1971-2000 (present) vs. 2071-2100 (future)]; 326-369 (present); 685-936 (future). | B. 1 degree [1971-2000 (present) vs. 2071-2100 (future)] 346 (present); | C. 12 km [2002-2004 (present) versus 2094-2096 (future)] 372-377 (present 802 (future) | D. 12 km Ensemble spread | E. 1 degree [1971-2000 (present) vs.2071-2100 (future)]; 326-369 (present); 685-936 (future). | F. 12 km [2002-2004 (present) versus 2094-2096 (future)] 372-377 (present); 802 (future) | G. 1.5 km [2002-2004 (present) versus 2094-2096 (future)] 371 (present); 800 (future) | H. 12 km Ensemble spread |
| **200m avg** | | | | | | | | |
| ΔTemp (°C) | 1.97 | 1.63 | 1.81, 2.55* | 1.38 to 2.24 | 2.21 | 1.95, 2.24* | 2.32, 2.62* | 1.55 to 2.35 |
| ΔOxygen (ml/l) | -0.19 | -0.52 | -0.62, -0.56* | -0.52 to -0.72 | -0.56 | -0.72, -0.66* | -0.61, -0.50* | -0.52 to -0.92 |
| ΔpCO₂ (µatm) | 401 | 492 | **759, 707*** | 682 to 836 | 527 | **923, 922*** | **721, 620*** | 780 to 1066 |
| ΔpH | -0.297 | -0.321 | -0.333, -0.322* | -0.310 to -0.357 | -0.332 | -0.315, -0.327* | -0.322, -0.311* | -0.278 to -0.352 |
| ΔΩ$_{arg}$ | -0.85 | -0.71 | -0.70, -0.68* | -0.65 to -0.75 | -0.62 | *-0.47, -0.53** | *-0.58, -0.62** | -0.41 to -0.53 |
| ΔΩ$_{cal}$ | -1.33 | -1.11 | -1.11, -1.08* | -1.03 to -1.19 | -0.99 | *-0.74, -0.85** | *-0.93, -0.98** | -0.65 to -0.83 |
| **Surface** | | | | | | | | |
| ΔTemp (°C) | 2.49 | 3.12 | 3.31, 3.27* | 2.57 to 4.05 | 3.15 | 3.30, 3.26* | 2.89, 2.86* | 2.42 to 4.18 |
| ΔOxygen (ml/l) | -0.23 | -0.32 | -0.35, -0.39* | -0.27 to -0.43 | -0.38 | -0.41, -0.41* | -0.40, -0.39* | -0.30 to -0.52 |
| ΔpCO₂ (µatm) | 379 | 392 | **435, 432*** | 433 to 437 | 365 | **429, 418*** | **352, 344*** | 424 to 434 |
| ΔpH | -0.309 | -0.319 | -0.286, -0.291* | -0.285 to -0.287 | -0.343 | -0.298, -0.297* | -0.274, -0.271* | -0.296 to -0.300 |
| ΔΩ$_{arg}$ | -0.98 | -0.96 | *-0.90, -0.85** | -0.86 to -0.94 | -0.76 | **-0.85, -0.85*** | **-0.76, -0.71*** | -0.82 to -0.88 |
| ΔΩ$_{cal}$ | -1.52 | -1.50 | *-1.42, -1.35** | -1.37 to -1.47 | -1.21 | **-1.35, -1.35*** | **-1.21, -1.14*** | -1.30 to -1.40 |
| **Bottom (<500m)** | | | | | | | | |
| ΔTemp (°C) | 1.98 | 1.65 | 1.84, 2.33* | 1.47 to 2.21 | 1.34 | **1.75, 1.90*** | **2.05, 2.35*** | 1.40 to 2.10 |
| ΔOxygen (ml/l) | -0.22 | -0.43 | -0.56, -0.64* | -0.37 to 0.75 | -0.63 | -0.66, -0.68* | -0.60, -0.60* | -0.40 to -0.92 |
| ΔpCO₂ (µatm) | 432 | 505 | **785, 883*** | 650 to 920 | 592 | **965, 1038*** | **840, 899*** | 776 to 1154 |
| ΔpH | -0.306 | -0.286 | -0.259, 0.318* | -0.228 to 0.290 | -0.333 | *-0.271, -0.292** | *-0.279, -0.308** | -0.230 to -0.312 |
| ΔΩ$_{arg}$ | -0.68 | -0.47 | *-0.42, -0.59** | -0.38 to 0.46 | -0.32 | -0.35, -0.38* | -0.43, **-0.50*** | -0.30 to -0.40 |
| ΔΩ$_{cal}$ | -1.06 | -0.74 | *-0.66, -0.93** | -0.59 to 0.73 | -0.50 | -0.56, -0.61* | -0.68, **-0.79*** | -0.48 to -0.64 |

**Fig. 1.**

[Figure]

**Fig. 2.**

---

## Author Comment (AC2) · 24 Nov 2020

We thank both reviewers for their insightful comments, which helped to improve the manuscript. Please find our point by point responses in the following proceeded by the word "RESPONSE". Please note that all of the figures and tables were updated as we re-ran the future projections. One of the ensemble members needed to be updated and so all of the runs associated with that ensemble member including the ensemble mean simulations were re-run. All of the numbers in the text, tables and figures were updating accordingly.

Anonymous Referee #2: I think that this manuscript is an extremely useful contribution

to the field, using a collection of projections at different spatial scales to identify how future biogeochemical change will be impacted by coastal processes operating at finer spatial scales. The analysis also benefits by its inclusion of a range of future climate projections, which allows for a quantification of the differences between models that is in excess of variations within the future climate projections beyond that of a difference from a mean future state. Below I make some general comments on the level of detail that the analysis includes – especially with regard to a more detailed quantification of why the local changes occurred and the specific effect of the coastal processes. I also make some specific comments on the discussion.

RESPONSE: We would like to thank the reviewer for their supportive and constructive approach to reviewing this manuscript. The resulting work is stronger and clearer as a result.

Reviewer Comments: General comments: (1) I think that this paper is a little more descriptive than I was expecting, or maybe hoping for. The aim of the paper is to describe how models that include higher spatial resolution of "coastal processes" lead to amplified or dampened responses to future change in comparison to coarser-scale models. The paper successfully describes how a model comparison can achieve this useful goal, but I think it falls a little short of clearly and specifically describing what processes and mechanisms lead to the amplified or dampened changes, based on the simulation results.

RESPONSE: In an effort to be more quantitative, we have added in two new tables to the supplement which detail the results of a new additional analysis.

Reviewer Comments: Let me try to give some specific examples: (a) Starting on line 442, there is a discussion of the differences in freshwater inputs and the related TA effects that reads "The TA changes are driven in part by the altered timing of the freshet in the N-CCS as well as the presence of a river plume in an upwelling regime. Freshwater in the region is known to be corrosive due to naturally low TA, which impacts the

buffering capacity of the surface waters. This result can be seen in the surface difference plots for pCO2 near the Columbia River plume (Fig. 4) and the surface TA change (Fig. 8). The 12-km projections include climatological freshwater fluxes as precipitation along the coastline instead of resolving river plumes like the 1.5-km projections, but despite these different freshwater parameterizations, both models indicate modification of carbon variables in the N-CCS Cascadia domain with different directions for different variables." What exactly is the reader to take away from this? How is TA actually changing on the NCCS (up or down), how does the altered representation of the freshwater effect TA and why is it important that a river discharges into an upwelling region?

RESPONSE: Figure 7 shows the differences in TA and DIC for various regions of the water column in both domains, but was improperly numbered in the submitted manuscript. Lines 383-384 in the manuscript detail the changes in TA: "TA increases in the future in the subsurface on the shelves of the CCS and even more so on the upper slope (Fig. 7). At the surface, it declines, and these two changes offset each other in the depth-averaged 200 meter change" In addition, we added the TA AND DIC changes, after the same formatting used in Table 1, to a new table, Table S1. We also added a plot of the freshwater forcing from the base and future runs from the 1.5 km simulations to the supplement as well. This figure details the discharge and the TA values associated with the rivers in the region is a constant (900 mmol/m3 ). The river water is lower in alkalinity than the oceanic end members.

Reviewer Comments: These questions are not addressed, and the paragraph ends with "both models indicate modification of carbon variables in the N-CCS Cascadia domain with different directions for different variables." It is not clear what "different directions for different variables" actually refers to – pH? TA? One goes up and the other down? And how does the freshwater control this specifically?

RESPONSE: Yes, that's right – one carbon parameter is amplified and another dampened and another is not modified at all. Lines 473-474:" While the freshwater amplifies

the global rate of change for the surface pCO2 and $\Omega$, over the entire water column (200 m average), the pH change is dampened" This is not solely because of freshwater in all regions of the water column, only at the surface. In an attempt to clarify this point, we include two new supplementary tables below. The first includes nutrients, TA, and DIC values similar to the format from Table 1 from the model runs. The second table (Table S2) showcases the R2 between the anomalies in Table 1 and either Table 1 or Table S1 with values >0.5 highlighted in grey.

Lines 475-481:" In our regional downscaled simulations, the change in temperature and TA act together to offset the increase in the DIC signal in the coastal upwelling regime (Fig. 7), and for pH these changes offset each other in the upper 200 m of the water column. The global models show very little change in TA in the region. While the downscaled bottom TA change is small (20-50 mmol/m3, Fig. 7), this amounts to an increase in pH of 0.07-0.18 and an increase in $\Omega$ of 0.15-0.21 - enough to offset 40-60% of the reduction in $\Omega$ due to increased atmospheric CO2 concentrations. At the bottom, the increase in TA modifies the projected pH change in the N-CCS by reducing it. "

We examined the relationship between the carbon variables and different forcing variables (just using R2). This is done using each row in Table 1 for the entire CCS region. The results of this exercise indicate that at the surface - DIC change matters, and at the bottom, changes in DIC and nutrient content changes have the highest correlation. The DIC is likely representative of the change from the RCP used, although it is lower in the downscaled simulations than the 1 degree models at the surface indicating biological production may also be playing a role (Table S1). At the bottom DIC is greater in the downscaled simulations than the 1 degree models (Table S1). Over the 200 m water column, TA is and nutrients are important. The TA decreases in all the projections but the decrease in the downscaled simulations is an order of magnitude less than in the 1 degree models (Table S1). A dampened reduction in TA is likely due again to the changes in the biological metabolism or benthic pelagic coupling alongside the sedimentary processes in the region - all of which can act as a source of TA on shelves.

If we take a specific example of $\Omega$, the $\Omega$ changes are correlated with nutrient changes, in addition to the DIC changes that result from the emissions scenario forcing. As a result, bottom $\Omega$ on the shelf is modified by the TA generation organic matter remineralization and via sedimentary processes. Freshwater plays a role at the surface, which you can see on the map of the changes in pCO2 at the surface in Figure 7. This occurs because the TA of the river water is different, and freshwater impacts the solubility of gases as well. DIC is correlated with $\Omega$ at the surface as well.

These results further bolster that the changes in different regions of the water column are driven by different mechanism - surface is more RCP, bottom is more benthic pelagic coupling and sedimentary processing on top of this, and the water column is a combination of these. The 1 degree models do not resolve the shelf, and thus the sedimentary processes that seem likely to play an important role in this coastal setting.

Reviewer comments: The text in the next paragraph beginning on Line 451 gives a specific example, so perhaps this example can be clearly wrapped into the paragraph before it starting on line 442, especially if that paragraph is more explicit about the nature of the climatological freshwater input versus the river plume, what specific effect on TA this change brings, perhaps describing what exactly the results in Fig 4 and 8 illustrate to support this conclusion.

RESPONSE: Figure 4 and now Figure 7 (formerly 8) both highlight the region in the N-CCS around the Columbia River plume in the surface fields in terms of the pCO2, TA, and DIC fields, but not much in the surface pH fields. This supports the conclusion that different carbon parameters respond differently to coastal processes and large scale climate forcing variables as a function of depth in the water column (proximity to surface freshwater fluxes and bottom sedimentary fluxes).

(b) Starting on line 485, downwelling is raised as a factor in modulating the response of

the N-CCS to future change, especially the fact that it may not change in the future. But it is not clear how the specific details that are subsequently mentioned, namely winter mixing, higher hypoxia, and seasonally persistent corrosiveness, relate to downwelling (or upwelling). Perhaps this might be obvious to a reader with detailed knowledge of the region, or that one is expected to assume how upwelling or downwelling might modulate the coastal response to climate change, but I think it needs to be more clearly organized.

RESPONSE: Thank you for helping us clarify this point. Currently the text states on line 517: "The fall transition and winter mixing re-oxygenates shelf waters (Siedlecki et al., 2015), and this pattern continues into the future. Hypoxia will continue to exist in the summer months, but for a longer period of the summer. Corrosive, low pH conditions, however, will occur year-round. The fraction of the year during which bottom water on the shelf is corrosive ($\Omega$ <1) or low pH (pH <7.65) nearly doubles. This asymmetry has been identified previously in the Salish Sea (Ianson et al., 2016), and exists because of the difference in equilibration timescales at the surface for oxygen and carbon dioxide" During the summer upwelling season, oxygen declines on the shelf over the season as organic material and respiration increases on the shelf. Coincident with that decline, carbon parameters are impacted in kind (e.g. pH and $\Omega$ decreases, pCO2 increases; Siedlecki et al. 2016; Feely et al. 2018). The dramatic fall transition on the Washington and Oregon shelves is observed in the modern ocean (Adams et al. 2013; Connolly et al. 2011; Hales et al. 2006; Siedlecki et al. 2015), fully quantified by an oxygen budget in Siedlecki et al. (2015), and is projected to continue in the future projections examined here. As the gas exchange propensity differs between oxygen and carbon dioxide, the transition impacts carbon dioxide differently and less dramatically. This result has been clearly shown and described further in the Salish Sea (Ianson et al. 2016). Some of these details have been added to the text surrounding this point to clarify for audiences unfamiliar with the region.

Reviewer Comments: Furthermore, the paragraph begins with stating how the shelves

and the N-CCS are projected to have greater change, but I don't think the association of these greater changes with upwelling/downwelling is clearly articulated in the paragraph. I also don't see how a specific mechanism is quantitatively related to the greater change based upon the results of the simulations the authors ran, and what the processes were in the better-resolved models that represented the process well enough to generate these changes.

RESPONSE: While these downscaled models likely represent upwelling better than their 1 degree counterparts as evidenced by their historical evaluations, the upwelling/downwelling intensity is not the source of the projected changes as shown by the CUTI and BEUTI indices not changes and as stated on lines 361-362:" Both of these measures suggest that the upwelling is not intensified in our projected future, despite the slight increase in winds. This result is consistent with the results of Howard et al. (2020), where increased stratification in the future simulations impeded increases in upwelling intensity." Our in water upwelling intensity measures take stratification into account.

Instead, the changes in the future are a result of the influence of freshwater at the land/ocean boundary, as well as the physical presence of the shelf bottom and the biogeochemical feedbacks associated with the sediment/water interface being resolved. We clarified this point by adding in an additional table (S2) to highlight the important correlations for the modified carbon variables – which differed depending on the portion of the water column which was the focus.

Reviewer Comments: (2) There are many places in the manuscript where statements like this are made: Line 362: "These changes in these depth ranges contribute to the results for the carbon variables in Table 1, impacting different carbon variables differently." These statements are too vague to be helpful, and in the case of this specific sentence, I expected the authors to then elaborate on what variables were different, how they were different, and where they were different, but that is not really achieved in the following sentences. This may sound picky, but I encourage the authors

to examine these types of statements and see how they can make them more specific, more informative, and more quantitative.

RESPONSE: We have tried to make these vague sentences more specific by adding in the results of a correlation analysis to a new supplemental table as well as specific examples to these sentences.

Reviewer Comments: (3) I would like to see the authors try and articulate some clear and specific conclusions of the paper. I understand their main point that resolving coastal processes matters for future projections, but there are places where the specifics of how they "matter" for the CCS for a given variable in the future could be more clearly stated. For example, in the concluding paragraph, it is written "Changes in pCO2 concentrations, âĎę, and pH are modified in the downscaled projections relative to the projected global simulation, suggesting downscaled projections are necessary to more accurately project future conditions of these variables." So, how are they modified? How do you think carbonate chemistry will be different in the CCS now that you have more resolved models, in contrast to what the global models say? More OA? Less OA? More seasonally variable OA?

RESPONSE: The downscaled models include coastal freshwater and sediment fluxes that aren't included in 1-degree simulations, and these fluxes seem large enough to explain substantial differences from the global models. Carbon variables are modified differently for different carbon variables in different portions of the water column as well as spatially within the CCS. Specifically, the change in $\Omega$ is modified, as this variable is more influenced by nutrient cycling and TA changes (Table S2). As a result, bottom $\Omega$ on the shelf is modified by the TA generation via sedimentary processes and increased benthic pelagic coupling that the presence of a bottom provides, while pH is not. "Future changes in pCO2 and surface $\Omega$ are amplified while changes in pH and upper 200 meter $\Omega$ are dampened relative to the projected change in global models" Ocean acidification trends in the region as measured by pH would match that set by the 1 degree models for the region. If you look at one of the other ocean acidification

parameters, or trends in another parameter like $\Omega$, depending on where in the CCS you were looking, you would observe a dampened rate of change in the S-CCS and an amplified rate of change in the N-CCS at the surface. pCO2 trends are amplified everywhere in the CCS.

Reviewer Comments: Specific comments: Line 22: The abstract sentence that begins with "These processes. . ." is a little confusing. Are the waters just generally low in oxygen and nutrients, or are the oxygen and nutrient concentrations in those waters projected to change, and thus work in concert with solubility change to alter future conditions? I think a small edit to the sentence will help clarify this.

RESPONSE: We think you mean this sentence:" These projected changes are consistent with source waters lower in oxygen, higher in nutrients, and in combination with solubility-driven changes, altered future upwelled waters in the CCS. " We meant both – that the source waters are generally low in o2 and high in nutrients, but the content is projected to change. In concert with solubility driven changes produces the projected changes. We have modified this sentence accordingly: "These projected changes are consistent with continued reduction in source waters oxygen, increase in source water nutrients, and, combined with solubility-driven changes, altered future upwelled source waters in the CCS. "

Reviewer Comments: Line 24: "coastal process resolving projections" is a mouthful and unclear, (and I understand word limits), but how about "projections that resolve coastal processes"?

RESPONSE: Thank you for your constructive suggestion. So modified.

Reviewer Comments: Line 41-42: So is the SST decline from Lima and Wethey 2012 predicted from a model? Or from a global-scale analysis that may not include local observations? It is not clear why this would differ from the Chavez record. Please add a sentence that describes why these two records give contrasting results.

RESPONSE: The L&W 2012 reference relied on NOAA OI $\frac{1}{4}$ degree daily SST data which relies on satellite data that has been quality controlled using in situ data sets from ICOADS. It is beyond the scope of this work to fully evaluate why the difference between these results, or even if the Chavez record was included in this data product, so the sentence has been removed.

Reviewer Comments: Line 83-87: This text seems out of place here, and perhaps should be moved to the section of the methods where you describe the three models.

RESPONSE: Thank you for your constructive suggestion. So modified.

Reviewer Comments: Line 125: What do you specifically mean by "spatially-weighted" means? Why calculate them?

RESPONSE: Different model cells have different volumes, and the average of the domain is not the simple average of every cell. Thus, the averages for each model needs to be weighted by the volume in that area specific to each grid.

Reviewer Comments: Line 126: "mode" should I think be "model"

RESPONSE: Thank you for catching this. So modified.

Reviewer Comments: Line 155: It is not clear that it is clear what "centennial trend addition" means here. Are there clear trends in these boundaries? Where do they come from?

RESPONSE: They come from the 100 year climate anomalies from the global models/1 degree models, from each ESM include climatological anomalies in salinity, temperature, nutrient and oxygen concentrations, and carbonate system parameters (Howard et al. 2020). These are added to the historical boundary conditions used to run the historical simulations. The text describing these methods has been clarified in the methods.

Reviewer Comments: Line 163: the text ". . ..,2007 with a one year. . .." just reads

awkwardly and is confusing. Was the year 2007 run for a year and then compared to observations from 2007?

RESPONSE: Thank you for your constructive suggestion. We modified the test to read:" To ensure no biogeochemical model drift between the nested 1.5-km simulation and the 12-km simulation, after one year of spin up, a simulation of 2007 was compared against observations from the region (Fig. 2)"

Reviewer Comments: Line 168: You make the argument, perhaps fairly, that any bias in the simulation generated by the configurations here will be the same in the two periods you compare. It would be more convincing, and help the reader if the reason for this bias was identified. Can this be evaluated? Is it simply a bias in the forcing?

RESPONSE: Because the "delta" forcing method was used, the climatological change over the 100 year ESM projections was added to the regional model forcing. Because the bias does indeed result from the forcing, it follows that with this delta method would result in shared (canceling) biases in both periods. e.g., the bias in the hindcast conditions can be expressed as Bias_h=ESM_h-Downscaling_h. The future conditions have Bias_f=ESM_f-(Downscaling_h + ESM_f –ESM_h) = Downscaling_h-ESM_h. Thus by construction the bias is identical in the two time periods. We have clarified the methods surrounding the delta method in hopes that this will clarify the text surrounding this bias as well.

Reviewer Comments: Line 186: Can you add a brief rationale for why you did not, for consistency sake, use the same 3-year future timeframe to compare all of the three models (used 30 years for coarse-scale model)? It seems unnecessary to add in this potential bias, but if bias is not an issue or there is another rationale for using all 30 years of the coarse scale run, please describe.

RESPONSE: This confusion has hopefully been clarified with the additional information in Table 1 regarding the consistent time windows compared between model simulations. While these additions do emphasize these differences a bit more, as reviewer 1

suggests, the differences in time do not account for the modification we find for the carbon variables. The difference between the carbon dioxide in the present (26 ppm) and future (112-136 ppm) between the 1 degree model and the 12 or 1.5 km simulations is not large enough to account for the modification we see in the runs. pCO2 from the 12km simulations in the upper 200 meters, for example, is 261 ppm greater than the 1 degree simulations project for the entire CCS. The modifications highlighted in Table 1 far exceed this difference in atmospheric carbon dioxide between the time periods. It is the 100 year anomalies that are shared across all analyses, and it is those differences that are being analyzed, not the absolute values of variables in any one particular year

Reviewer Comments: Line 197: I think it would help to state the stressor variables parenthetically in this sentence.

RESPONSE: Thank you for your constructive suggestion. So modified.

Reviewer Comments: Line 346: Perhaps there is a convention in the language of this upwelling system that I am unfamiliar with, but why is higher NO3 associated with more O2 drawdown? Is it because the NO3 increase is a tracer of upwelling that can be linked to O2 source water that has a certain, lower O2 signature? Or is this NO3 assumed to be taken up by phytoplankton growth and subsequently used to drawdown O2 at depth?

RESPONSE: It is both. Higher nutrients in the source water corresponds to lower oxygen in the source water but increased nutrients in the source waters drives productivity higher. Higher production leads to greater nutrient trapping and more oxygen drawdown on the shelf. The results from Howard et al. (2020) indicate that production within the CCS is probably not greater in the future projections for the region suggesting that the net balance of oxygen and nitrate fluxes are unchanged and larger nitrate changes must be balanced by larger oxygen changes.

Reviewer Comments: Line 374: The word "modified" is used to describe the relative pH changes, but the text that follows seems to consistently describe dampening. Can't

you replace "modified" with "dampening" to be clearer?

RESPONSE: While pH is dampened, pCO2 is amplified. Because different variables have a different response, we chose to use modified to encompass both.

Reviewer Comments: Line 426: It is unclear if "values" refers to the delta pH or the mean pH when comparing to the other studies.

RESPONSE:Thank you for your constructive suggestion. So modified to include pH in front of values.

Reviewer Comments: Line 428: Please clarify that you are describing your "downscaled projections" here, and not those of Dunne

RESPONSE: Thank you for your constructive suggestion. So modified to clarify that we refer to our projections.

Reviewer Comments: Line 458: Here is a place where some specific model details might provide quantitative information to support the discussion. Denitrification is raised as a process that can affect TA, but the actual differences in denitrification (and its TA effect) in the model simulations are not shown. I understand that there is a limited amount of information to be shown in any paper, but this discussion would be more compelling if the potential denitrification change was reported. Maybe it is a weak effect, maybe strong, and it would be helpful to know.

RESPONSE: TA differences are shown in Figure 7 (now properly numbered – was Figure 8 in the version you read). We have now added a supplementary table with the TA changes specifically quantified. This difference, however, was not put in context of the global model changes. We have now added this into the table 1 and text. As you can see the downscaled simulations are projected to decline in TA but less so than the 1 degree models for the same region. We were unable to report the denitrification fluxes in particular as they were not written out for any of the runs analyzed here. Instead, we calculated N* for the simulations we could (for those that include phosphate information). The results were not included because they were not conclusive. N* increases in all the simulations in the future, but less so in the downscaled simulations. N* is also highly correlated to the TA and $\Omega$ changes also provided in the new supplementary Table. We interpret these results to be consistent with our previous findings but realize that we cannot distinguish between changes in transport of the N* signal into the domain, respiration, and denitrification impacts on the water column using this approach, so we altered the language in the discussion to reflect this and look forward to performing future dedicated experiments to address this question. Those experiments are beyond the scope of this inquiry.

Reviewer Comments: Figure 4: Have you considered plotting these deltas on a percentage scale? I understand why they are plotted the way they are, to show absolute changes, but the scales are different (necessarily?) across the depths and this makes it a little harder to compare them, if one wanted.

RESPONSE: We have considered this but have chosen to first plot everything on the same color scale as per Reviewer 1's request. In order for the percent difference plots to be meaningful, we would also have to include base condition maps for all variables and all depths, which would add additional figures to the paper. It is not clear that adding these plots would alter any of the conclusions presented either.

Please also note the supplement to this comment:
https://bg.copernicus.org/preprints/bg-2020-279/bg-2020-279-AC2-supplement.pdf

**Supplement:**

**Supplemental material:**
**Coastal processes modify projections of some climate-driven stressors in the California Current System**

**S.A. Siedlecki[1*], D. Pilcher[2], E.M. Howard[3], C. Deutsch[3], P. MacCready[3], E.L. Norton[2], H. Frenzel[3], J. Newton[4], R.A. Feely[5], S.R. Alin[5], and T. Klinger[6]**

[1]Department of Marine Sciences, University of Connecticut, 1080 Shennecossett Road, Groton, CT 06340, USA
[2]CICOES, University of Washington, 3737 Brooklyn Ave NE, Seattle, WA 98195, USA.
[3]School of Oceanography, University of Washington, 1503 NE Boat Street, Seattle, WA 98195, USA.
[4]APL, University of Washington, 1013 NE 40th Street, Seattle, WA 98105, USA.
[5]NOAA Pacific Marine Environmental Lab (PMEL), 7600 Sand Point Way NE, Seattle, WA 98115, USA.
[6]School of Marine Environment and Affairs, University of Washington, 3707 Brooklyn Ave NE, Seattle, WA 98105, USA.

Corresponding author: Samantha Siedlecki (samantha.siedlecki@uconn.edu)

| | A. CCS (1 degree) | B. CCS (12 km) | C. N-CCS (1 degree) | D. N-CCS (12km) | E. N-CCS (1.5km) |
|---|---|---|---|---|---|
| **TA surf** | -20.41 | -7.88 -5.88* | -26.01 | -7.40 -6.29* | -7.46 -10.53* |
| **TA 200m** | -8.90 | 0.68 -3.10* | -7.72 | 1.91 -1.0* | -3.05 -8.22* |
| **TA bot** | -10.38 | -0.03 -1.90* | -7.70 | 1.00 0.28* | -1.66 -4.17* |
| | | | | | |
| **DIC surf** | 101.44 | 93.32 91.65* | 80.52 | 89.93 91.26* | 80.98 73.57* |
| **DIC 200m** | 96.82 | 103.24 94.35* | 78.04 | 89.49 91.74* | 87.37 80.93* |
| **DIC bot** | 66.33 | 75.25 92.51* | 69.63 | 79.92 83.93* | 77.66 83.71* |
| | | | | | |
| **PO4 surf** | -0.0561 | -0.0727 -0.0754* | -0.0403 | -0.0732 -0.0802* | N/A |
| **PO4 200m** | 0.0655 | 0.0273 -0.0180* | 0.0191 | 0.0337 0.0062* | N/A |
| **PO4 bot** | -0.0064 | 0.0112 0.0096* | 0.0131 | 0.0397 0.0342 | N/A |
| | | | | | |
| **NO3 surf** | -0.3216 | -0.1869 -0.2831* | -0.3449 | -0.3247 -0.3678* | |
| **NO3 200m** | 1.1383 | 1.4705 0.5535* | 0.8318 | 1.2999 0.8426* | |
| **NO3 bot** | 0.6392 | 0.6374 0.7061* | 0.6833 | 0.9136 0.8799* | |

**Table S1: Annual average differences between climate stressor variables in the future and the base/modern conditions for the 1 degree, 12-km, and 1.5-km projections over different regions of the water column (200 m averaged, surface, and bottom < 500 m). Column A includes the global (1 degree) ensemble average difference for the CCS region followed by, in column B, the CCS wide difference from the 12km downscaled results. The N-CCS region results span columns C-E in this table. The next three columns detail the differences in the Cascadia domain for the global ensemble average (column C), the 12-km (column D) and 1.5-km downscaled projections (column E). Within the downscaled simulation, values are also provided just on the shelf (< 200 meter isobath) and denoted with an asterisk (*) next to the number.**

| | 200m avg | Surface | Bottom (<500m) |
|---|---|---|---|
| $\Delta pCO_2$ (µatm) & $\Delta TA$ | 0.94 | 0.19 | 0.92 |
| $\Delta pCO_2$ (µatm)& $\Delta$ DIC | 0.04 | 0.34 | 0.99 |
| $\Delta pCO_2$ (µatm) & $\Delta NO_3$ | 0.47 | 0.35 | 0.62 |
| $\Delta pCO_2$ (µatm) & $\Delta$ T | 0.01 | 0.81 | 0.36 |
| $\Delta pH$ & $\Delta TA$ | 0.04 | 0.74 | 0.59, p=0.20 |
| $\Delta pH$ & $\Delta$ DIC | 0.001 | 0.81 | 0.28 |
| $\Delta pH$ & $\Delta NO_3$ | 0.03 | 0.56, p=0.3 | 0.003 |
| $\Delta pH$ & $\Delta T$ | 0.01 | -0.19 | 0.03 |
| $\Delta \Omega$ & TA | 0.27 | -0.0009 | 0.66, p=0.21 |
| $\Delta \Omega$ & $\Delta$ DIC | 0.25 | 0.99 | 0.71 |
| $\Delta \Omega$ & $NO_3$ | 0.0002 | 0.14 | 0.95 |
| $\Delta \Omega$ & T | 0.24 | 0.22 | 0.002 |

**Table S2: We examine the relationship between the carbon variables that are modified in the downscaled simulations and the other variables representative of different processes. The $R^2$ between the anomalies in Table 1 and either Table 1 or Table S1 with values >0.5 highlighted in grey. All p values were less than 0.05 indicating significant correlations.**

[Figure]

**Figure S1.** Global changes from the CMIP5 ensemble average for the CCS region - future minus base conditions (a) temperature (deg C) (b) $O_2$ (ml/l) (c) $pCO_2$ (µatm) (d) pH.

[Figure]

**Figure S2.** Freshwater discharge forcing for the 1.5 km simulation in the present and future simulations as described in the methods.

---

## Author Response (AR2)

We would like to thank the reviewers for their time and efforts in making our paper clearer. We have addressed all the minor concerns raised by the reports at this stage. In addition we updated the data availability section to better reflect where the model data presented here will be accessible.

Report #1:

The authors addressed my earlier comments and made the manuscript clearer. Unless the other reviewer(s) identify new additional problems, I recommend acceptation of the manuscript, conditional to the following minor changes. (1) The new Table S1 (in Supplement) completely lacks units. Please add units in Table S1 for all variables.

**Response: Units have now been added to the caption for Table S1. Thank you for pointing out this issue to us as again it helps make the paper clearer.**

(2) The caption for the new Table S2 (in Supplement) needs clarifications. How are the correlations computed? Is it a spatial correlation between, e.g., surface pCO2 anomaly and surface temperature anomaly? The reader shouldn't have to guess these things!

*Response:*
The caption text used to read:
**The $R^2$ between the anomalies in Table 1 and either Table 1 or Table S1 with values >0.5 highlighted in grey.**

We have now altered this to read:
**The $R^2$ between the anomalies in Table 1 and either Table 1 or Table S1 with values >0.5 highlighted in grey. For example for the correlations between pco2 and TA over the 200 m avg – the correlations are between the $\Delta$pCO2 in the 200 m avg section from table 1 for the entire CCS region (column C) from the downscaled models and the TA from column B from Table S1 for the 200m row.**

Report #2:
I identified a few minor language errors in the new text that should be fixed.

Response:

Ok, without specific direction from the reviewer, we followed your advice and focused on the new text. Upon another read through of the newer text, we found some minor edits and edited the document accordingly.